**Investigation of processes controlling summertime gaseous elemental mercury**
**oxidation at mid-latitudinal marine, coastal, and inland Sites**
Z. Ye[1], H. Mao[1], C.-J. Lin[2,3], and S. Y. Kim[4]
[1] Department of Chemistry, State University of New York College of Environmental Science and
Forestry, Syracuse, NY, 13210, USA
[2] Center for Advances in Water and Air Quality, Lamar University, Beaumont, TX, 77710, USA
[3] Department of Civil and Environmental Engineering, Lamar University, Beaumont, TX, 77710,
USA
[4] R&D Program Evaluation Division Office of National Evaluation and Analysis Korea Institute
of S&T Evaluation and Planning (KISTEP), Seoul, South Korea
Received: 13 October 2015 – Accepted: 23 October 2015 – Published: 15 January 2016
Correspondence to: Z. Ye (zye01@syr.edu)

**Abstract**

A box model incorporating a state-of-the-art chemical mechanism for atmospheric mercury (Hg) cycling was developed to investigate oxidation of gaseous elemental mercury (GEM) at three locations in the northeastern United States: Appledore Island (marine), Thompson Farm (coastal, rural), and Pack Monadnock (inland, rural, elevated). The chemical mechanism in this box model included the most up-to-date Hg and halogen chemistry. As a result, the box model was able to simulate reasonably the observed diurnal cycles of gaseous oxidized mercury (GOM) and chemical speciation bearing distinct differences between the three sites. In agreement with observations, simulated GOM diurnal cycles at AI and TF showed significant daytime peaks in the afternoon and nighttime minimums compared to flat GOM diurnal cycles at PM. Moreover, and significant differences in magnitude of GOM diurnal amplitude (AI>TF>PM) were captured in modeled results. At the coastal and inland sites, GEM oxidation was predominated by $O_3$ and OH, contributing 80–99% of total GOM production during daytime. $H_2O_2$ initiated GEM oxidation was significant (~33% of the total GOM) at the inland site during nighttime. In the marine boundary layer (MBL) atmosphere, Br and BrO became dominant GEM oxidants with mixing ratios reaching 0.1 and 1 pptv, respectively, contributing ~70% of the total GOM production during mid-day, while $O_3$ dominated GEM oxidation (50–90% of GOM production) over the remaining day when Br and BrO mixing ratios were diminished. The majority of HgBr produced from GEM+Br was oxidized by $NO_2$ and $HO_2$ to form brominated GOM species. Relative humidity and products of the $CH_3O_2$+BrO reaction possibly affected significantly the mixing ratios of Br or BrO radicals and subsequently GOM formation. Gas-particle partitioning could be potentially important in the production of GOM as well as Br and BrO at the marine site.

# 1 Introduction

Mercury (Hg) is a toxic pollutant found globally in air, natural waters, and soils. The health concern of Hg arises from the neurotoxic organic form, methyl mercury (MeHg), in the aquatic environments (Mason et al., 2006; Miller et al., 2007; Rolfhus et al., 2003). The high bioaccumulation and biomagnification of MeHg lead to human exposure through the consumption of seafood (Clarkson, 1994). Deposition of atmospheric Hg is one of the most important sources of aquatic Hg.

In the atmosphere, Hg exists in three forms: gaseous elemental mercury (GEM), gaseous oxidized mercury (GOM), and particulate bound mercury (PBM). The majority of atmospheric Hg is GEM, comprising > 95% of total gaseous mercury (TGM=GEM+GOM). The 0.8–1.7 years atmospheric lifetime of GEM is conducive to long range transport of Hg as a global pollutant (Bergan et al., 1999; Bergan and Rodhe, 2001; Holmes et al., 2006; Lin and Pehkonen, 1999; Schroeder and Munthe, 1998; Selin et al., 2007). In contrast, GOM and PBM are relatively short-lived and subject to wet deposition and stronger dry deposition than GEM due to their high solubility in water and low vapor pressure. GOM in the atmosphere can be produced from oxidation of GEM, released directly from anthropogenic emissions, and transformed from PBM. In remote regions, in-situ GOM production may be the major source of GOM (Weiss-Penzias et al., 2003; Poissant et al., 2004; Mao and Talbot, 2012) considering its short lifetime.

Chemical speciation of atmospheric Hg is essential to understand its geochemical cycle. Theoretical and experimental studies suggested that the main oxidants of GEM in the atmosphere are ozone ($O_3$), hydroxyl radical (OH), atomic bromine (Br), bromine monoxide (BrO), hydrogen peroxide ($H_2O_2$), and atomic chlorine (Cl), yielding GOM species of HgO, HgBrO, HgBr, $Hg(OH)_2$, HgCl, and through further reaction to other mercury halides (Ariya et al., 2015; Dibble

et al., 2012; Lin and Pehkonen, 1999). Although efforts have been made to investigate the relative
importance of these oxidants for GEM oxidation in the troposphere, it is still not well understood.
In the terrestrial environment, it was suggested that the oxidation of GEM was primarily by $O_3$ and
OH radicals (Shon et al., 2005; Sillman et al., 2007). The speciation and quantification of GEM +
$O_3$ product(s) still remain unknown and debatable (Ariya et al., 2015; Gustin et al., 2013; Rutter
et al., 2012). An experimental study by Pal and Ariya (2004b) measured 1% of HgO produced by
GEM + $O_3$ on an aerosol filter. Snider et al. (2008) showed HgO(s) production in their kinetic and
product study. Schroeder et al. (1998) suggested HgO would not exist as an isolated molecule in
gas phase but could be deposited to and retained by manifold given a decomposition temperature
of +500 °C. However, the GEM + $O_3$ reaction and decomposition temperature (Schroeder et al.,
1998) could also be impacted by the presence of other ambient gases (Snider et al., 2008; Gustin
et al., 2013; Seigneur et al., 1994). A recent study of Huang et al. (2013) observed gas-phase HgO
using nylon and cation exchange membranes. The reaction of GEM+OH has been subject to debate
between theoretical and experimental studies, as no mechanism consistent with thermochemistry
has been proposed (Ariya et al., 2015; Pal and Ariya, 2004a; Subir et al., 2011). Measurement
studies on GOM in polar regions (Simpson et al., 2007; Steffen et al., 2008) and sub-tropical MBL
(Laurier et al., 2003; Laurier and Mason, 2007; Obrist et al., 2011) as well as atmospheric modeling
studies on mercury cycling (Holmes et al., 2009, 2010; Kim et al., 2010; Lindberg et al., 2002;
Obrist et al., 2011; Soerensen et al., 2010; Toyota et al., 2014; Wang et al., 2014; Xie et al., 2008)
have suggested Br as an important oxidant of GEM. The major source of atmospheric Br was
suggested to be produced photolytically from Br-containing compounds and through the Br/BrO
cycle involving tropospheric $O_3$ (Saiz-Lopez and Glasow, 2012; Simpson et al., 2015).

GOM concentrations and speciation could be impacted by meteorological conditions and

chemical conditions in different environments. High solubility of GOM species, possible phase

partitioning of HgO as discussed above could all be the reasons causing varying GOM speciation

at different locations. For instance, the aerosol type, size distribution, and chemical composition

varied largely between the MBL site and inland sites, which may lead to different gas-particle

partitioning rates of GOM species.

Hg chemistry in the MBL, the lowest part of the troposphere in direct contact with the sea

surface, has global importance as approximately 70% of the earth's surface is covered by oceans

(Glasow et al., 2002). Hg in the MBL cycles differently from in coastal or inland areas. However,

contemporary models are not able to reproduce GOM observations temporally and spatially due

to knowledge gaps in Hg science, simplified model assumptions, and uncertainties of

measurements (Ariya et al., 2015; Lin et al., 2006). In the polar region, bromine radicals were

identified as the primary cause of the Arctic mercury depletion events (AMDE) (Kim et al., 2010;

Lindberg et al., 2002; Toyota et al., 2014; Xie et al., 2008). In the MBL outside Polar Regions,

due to lower mixing ratios of atmospheric halogen radicals, often lower than the detection limit,

mechanisms for GOM production were more controversial than in Polar Regions. Using a box

model, Hedgecock et al. (Hedgecock et al., 2003; Hedgecock and Pirrone, 2004, 2005) suggested

that $O_3$ was a dominant GEM oxidant in the MBL at mid-latitudes in the Mediterranean region,

and that the GEM+$O_3$ reaction may form solid products. However, the reaction kinetics in their

model were out-of-date with limited halogen chemistry, and fixed emissions used in the model

oversimplified the source terms. Holmes et al. (2009) simulated that GEM oxidation by Br

comprised 35–60% of the GOM sources using BrO concentrations calculated at a photostationary

state from a prescribed distribution of Br mixing ratios. Additionally, a parameter was introduced

in the same study to account for entrainment of free tropospheric GOM into the MBL and the Br
mixing ratio was adjusted to capture the observed GOM diurnal trend, which could cause large
uncertainties in GOM simulations. Most recently, Wang et al. (2014) employed updated Hg
reactions together with bromine and iodine reactions, adopting the free tropospheric GOM
entrainment parameter from Holmes et al. (2009) for tropical MBL, and found Br to be a primary
GEM oxidant, but oxidation by Br or $O_3$/OH alone was unable to reproduce observed GOM
concentration. However, different GEM oxidants could be dominant in different environments, as
a result of the unique composition and concentration levels of in-situ oxidants those environments
may be characterized with.

In this study, we employed a state-of-the-art chemical mechanism that incorporates gas and

aqueous phase chemistry of Hg, $O_3$, and halogen to investigate the dynamics of GOM formation
under various atmospheric conditions in mid-latitude regions. The most up-to-date kinetics was
applied. Halogen radical mixing ratios (such as Br and BrO) were calculated using up-to-date
atmospheric halogen reactions. Clear sky days with calm wind conditions were selected, which are
mostly associated with strong atmospheric stability, to minimize the entrainment effect of free
tropospheric air and regional transport and hence no entrainment factor was included in this study.
Moreover, the initial GEM mixing ratios along with a list of compounds (Table 2) in the model
were obtained from observations in three different environments and were set to be constant during
simulations. Fixing the input concentrations of GEM among a number of other compounds (Table
2) as constants using observational data enabled a modeled chemical environment close to the real
atmospheric environment that is being studied.    Moreover, a box model simulates the
concentrations of short-lived compounds reaching an instantaneous chemical steady state, and for
the time scales of such instants, the chemicals such as GEM are long-lived enough to maintain a
constant level. In Section 2, the methods employed were laid out in detail. Section 3 presented
results of reasonably simulated differences between GOM diurnal cycles at the three locations that
were captured in measurement data, major GEM oxidants in the three environments, and a detailed
discussion of the sensitivity of physical parameters and important chemical reactions. Section 4
summarized the key findings and implications from this study.
## 2    Methods
### 2.1    Box Model Description
The Kinetic PreProcessor version 2.1 (Sandu and Sander, 2006) was utilized as the
framework of the box model (Hedgecock et al., 2003; Hedgecock and Pirrone, 2004, 2005). A
second order Rosenbrock method (Verwer et al., 1999) was applied to solve the coupled ordinary
differential equations. The box model used in this study was initially set up by Kim et al. (2010).
It was further improved in this study by incorporating the most up-to-date gas and aqueous phase
chemical mechanisms (Atkinson et al., 2004, 2008; Dibble et al., 2012; Sander et al., 2011) to the
model.
2.1.1    Reactions and kinetics
The box model has a total of 424 reactions: 276 gas-phase reactions (including Hg, halogen,
$O_3$, sulfate, and hydrocarbon reactions), 52 gas-water equilibriums, 28 aqueous equilibriums and
68 aqueous reactions. Most of these reactions and kinetic data were updated based on JPL Report
No. 17 (Sander et al., 2011), the halogen chemistry reviews of Atkinson et al. (2004, 2008), and
the references listed in Table 1. Photodissociation coefficients were calculated from the
Tropospheric Ultraviolet and Visible (TUV) Radiation Model (Madronich, 1993).
The most important improvements in chemistry are gas and aerosol phase of Hg and
halogen reactions. Gas-phase Hg reactions included in the box model are (Table 1):
1. Oxidation of GEM by $O_3$, OH, $H_2O_2$, Br, BrO, Cl, $Cl_2$, I (G1–8);
2. Reduction of HgBr and HgI to produce GEM (G9–11); and
3. Reactions of HgBr/HgCl with BrO, ClO, IO, $NO_2$, $HO_2$, and OH (G12–24) with kinetics
suggested by Dibble et al. (2012).
Aqueous Hg reactions include (Table S1):
1. Oxidations of Hg by $O_3$, OH, HOCl, and $ClO^-$, further oxidation of HgOH by $O_2$;
2. Reduction of $Hg^{2+}$ by $HO_2$, photolytic reduction of $Hg(OH)_2$ and S(IV)-mediated
reduction; and
3. Aqueous equilibria involving $HgSO_3$, $Hg(SO_3)_2^{2-}$, $HgOH^+$ and $Hg(OH)_2$. Gasphase
halogen reactions in the box model are mainly cycles of halogen radicals (Cl/Br/I and ClO/BrO/IO
radicals).
The Cl/Br/I radical cycles include photodissociation of $Cl_2/Br_2/I_2$, organic halides, and
other inorganic halides as sources, and oxidation reactions as sinks. The ClO/BrO/IO radical cycles
involve oxidation of Cl/Br/I radicals, photodissociation of $ClNO_2/ClONO_2/BrNO_2/BrONO_2$,
production from other halogen radicals, and sink reactions to calculate Cl/Br/I radicals or other
halides. Aqueous halogen reactions include reactions of $Br^-/Cl^-$ and reactions of aqueous BrCl,
HCl, HBr, HOCl, HOBr, $Cl_2$, and $Br_2$ species. The chemistry of halogen radicals, especially the
reaction cycles of Br and BrO radicals, could be important and should not be neglected or replaced
by simple approximation. Hence, the most up-to-date halogen chemistry from the literature was
included in our model.
2.1.2   Initial conditions and input data
Observations at three sites from the University of New Hampshire (UNH) AIRMAP
Observing Network (http://www.eos.unh.edu/observatories/data.shtml) were used: a marine site
located on Appledore Island (AI) at the Shoals Marine Lab, the Gulf of Maine (42.97°N, 70.62°W,
40 m a.s.l.), a coastal site located in Thompson Farm (TF) in Durham, NH (43.11°N, 70.95°W, 24
m a.s.l.) and 25 km away from the Gulf of Maine, and a forested 90 km inland site located on Pack
Monadnock (PM) in Miller State Park in Peterborough, NH (42.86°N, 71.88°W, 700 m a.s.l.) (Fig.
1). Hourly mean values of GEM, $O_3$, CO, NO, meteorological observations (i.e., temperature,
relative humidity, and solar radiation) at these three sites were used as initial input to the box model.
For species that were not measured, we set their initial concentrations as the values in similar
environments from the literature if available. Observations of GOM mixing ratios from the three
sites were utilized to evaluate the model performance. GEM and GOM data were collected using
the Tekran® 2537/1130/1135 speciation unit (Tekran Inc., Canada). For these three sites, the
instruments were first run and calibrated in the laboratory and then operated at the sites in a
consistent manner. GEM was measured at 5-min intervals and with a limit of detection (LOD) of
~5-10 ppqv (Mao et al., 2008), GOM was measured over a 2-h sampling period with a LOD of
~0.1 ppqv based on three times the standard deviation of the field blank values (Sigler et al., 2009;
Mao and Talbot, 2012). A custom-built refrigerator assembly and a canister of drierite was used
to cool and dry air streams before entering into the 1130 pump module, resulting in < 25% RH of
air streams (Sigler et al., 2009). Detailed information on these measurements can be found in Mao
and Talbot (2004; 2012), Talbot et al. (2005), Fischer et al. (2007), Mao et al. (2008), and Sigler
et al. (2009). Table 2 lists the input variables of the box model. The model's initial mixing ratios
of GEM, $O_3$, CO, and NO were obtained from observations and were set to be constant during
each 1h simulation. Br/Cl/I concentrations were all calculated from the model given initial
concentrations of 1 pptv (Finley et al., 2008; except for AI) for $Br_2$, $Cl_2$, and $I_2$ species. At AI, the
$Br_2$ initial concentration was set to be constant during simulations and used Saiz-Lopez et al.
(2006)'s values to constrain [BrO]. Detailed information can be found in Section 3.3.1. Dry
deposition flux was calculated using dry deposition velocity data derived from Zhang et al. (2009,
2012) and boundary layer height estimated from Mao and Talbot (2004). Other physical
parameters (i.e. Henry's constants, liquid water content, and aerosol radius) were used to simulate
the gas-particle partitioning process in the box model.
2.1.3 Gas-particle partitioning
An empirical expression was utilized to calculate particle size growth relative to its dry
radius ($r_{dry}$) (Lewis and Schwartz, 2006):
$$r = r_{dry}\frac{4}{3.7}\left(\frac{2-RH}{1-RH}\right)^{1/3},$$
(1)

where RH is the relative humidity, and r is the particle radius at RH.
Gas-particle partitioning was treated by mass transfer between droplets and air. The
dynamic mass transfer coefficient across the gas-aqueous interface was calculated using the
method developed by Schwartz (1986). The net mass flux ($F$, molecule cm$^{-3}$ s$^{-1}$) between the gas
and aqueous phase is given by
$$F = k_{mt} \times (L \times c_g - \frac{c_{aq}}{HRT}),$$
(2)

where L is the liquid water content ($m^3_{water}m^{-3}_{air}$), $k_{mt}$ is the mass transfer coefficient (s$^{-1}$), $c_g$ is the
gas phase concentration of the species (molecules cm$^{-3}$), $c_{aq}$ is the aqueous phase concentration of
species (molecules cm$^{-3}$), H is the Henry's constant of the species (M atm$^{-1}$), R is the universal gas
constant (atm L K$^{-1}$ mol$^{-1}$), and T is atmospheric temperature (K). $k_{mt}$ is calculated as follow:
$$k_{mt} = \left(\frac{r^2}{3D_g} + \frac{4r}{3\bar{v}\alpha}\right)^{-1},$$
(3)

$$\bar{v} = (8RT/M\pi)^{1/2},$$
(4)

where r is the particle radius (μm), $D_g$ is the diffusion coefficient ($m^2 s^{-1}$), $\bar{v}$ is the mean thermal
molecular velocity (m $s^{-1}$), α is the dimensionless accommodation coefficient, and M is the species
molecular weight (g $mol^{-1}$).

## 2.2   Case Selection

A total of 83 cases were examined to investigate the role of chemistry in Hg cycling in the

MBL, coastal, and inland environments. At the study sites, significant warm season declines of
GEM were observed with annual maximums in spring and minimums in autumn resulting in
seasonal amplitudes up to 100 ppqv at TF (Mao et al., 2008). The lost GEM during the warm
season most likely entered the ecosystem. Chemical transformation of GEM in warm seasons was
suspected to be one of the factors causing the observed seasonal decline in GEM. As such, this
study selected the cases representing summer days when chemical processes were most likely
dominant. To exclude the influence of wet deposition, we selected clear-sky conditions based on
the observed photodissociation rate constant of $NO_2$ ($jNO_2$) and solar radiation flux. To minimize
the influence of transport, cases with arithmetic daily mean wind speed higher than 75% percentile
of all summer days in studied years (> 1.3 m $s^{-1}$ at TF, > 6 m $s^{-1}$ at PM, and > 7 m $s^{-1}$ at AI) were
excluded. As a result, 50, 12, and 21 clear-sky days at AI (marine), TF (coastal), and PM (inland,
elevated), respectively, were selected from summers of 2007, 2008, and 2010. Since there was no
temperature data available for summer 2009 at TF, 2009 was not considered.

## 2.3   Backward Trajectory Model

The National Oceanic and Atmospheric Administration (NOAA) Hybrid Single Particle

Lagrangian Integrated Trajectory (HYSPLIT) trajectory model was used to identify source regions
of air masses at the three sites. The model runs were performed over twenty-four hours using the
NOAA NAM (Eta) Data Assimilation System (EDAS) data with a 40km×40km horizontal
resolution as input. Backward trajectories and trajectory clusters were calculated.
**2.4   Model Evaluation**
To evaluate the box model performance with observations, the following statistical
performance measures (Chang and Hanna, 2004; Hanna, 1988; Hanna et al., 1991, 1993), which
include the fractional bias (FB), the normalized mean square error (NMSE), the root mean square
error (RMSE), and the partition of NMSE due to systematic errors (NMSE$_s$) were used:
$$FB = \left(\overline{C_0} - \overline{C_p}\right)/0.5\left(\overline{C_0} + \overline{C_p}\right), \tag{5}$$

$$NMSE = \overline{\left(C_0 - C_p\right)^2}/\overline{C_0 C_p}, \tag{6}$$

$$RMSE = \sqrt{\overline{\left(C_0 - C_p\right)^2}}, \tag{7}$$

$$NMSE_s = 4FB^2/\left(4 - FB^2\right), \tag{8}$$

where $C_p$ is model predictions, $C_0$ is observations, overbar ($\bar{C}$) is average over the dataset.
**3   Results and Discussion**
**3.1   General characteristics in measured GOM and GEM**
In the selected 83 cases, atmospheric GOM and GEM mixing ratios varied greatly at the
three sites (Fig. 2). Mixing ratios of GOM varied over 0.03–87.79 ppqv at AI, 0.04–4.93 ppqv at
TF, and 0–0.65 ppqv at PM. At AI and TF, significant diurnal variation was observed with
afternoon maximums and nighttime minimums. At AI, GOM peaked at 10 ppqv over 14:00–16:00
EDT and was ~ 5 ppqv at night, well above the LOD (~ 0.1 ppqv, from Sigler et al., 2009, the
same LOD for the instruments at the three sites). At TF, GOM mixing ratios peaked at 0.75 ppqv
at 17:00 EDT and were below LOD at night, before 08:00 EDT. The GOM diurnal cycle at PM
was different from that at AI and TF. At PM, averaged GOM had higher mixing ratios at night and
in the early morning than in the afternoon. However, the median values showed afternoon peaks
and nighttime minimums. The difference between average and median GOM diurnal cycles was
driven by 3 cases that had abnormally high GOM mixing ratios (> 0.6 ppqv) at night or in the early
morning relative to the average GOM mixing ratio through the day (~ 0.1 ppqv).

Mixing ratios of GEM ranged over 65–231 ppqv at AI, 60–213 ppqv at TF, and 121– 231

ppqv at PM (Fig. 2). On average, GEM mixing ratios at PM were 8% higher than that at TF and
12% higher than that at AI. Unlike GOM, GEM diurnal cycles showed nearly flat patterns at AI
and PM, though slightly higher (~ 3 %) GEM mixing ratios at night than in the daytime were
observed at PM. In contrast, the average GEM diurnal cycle at TF showed an early morning (07:00
EDT) minimum (112 ppqv) and a daytime (13:00 EDT) maximum (153 ppqv).

The site differences of GOM and GEM diurnal cycles could be attributed to different

chemical environments, land surface types, and meteorological conditions. For example, the GEM
daily minimum at night and in the early morning at TF was likely caused by a strong net loss
dominated by dry deposition under nocturnal inversion (Mao et al., 2008; Mao and Talbot, 2012).
Nocturnal inversion also influenced the GEM and GOM diurnal cycles at PM, albeit differently
from at TF. The elevation of PM site is 700ma.s.l., above the nocturnal inversion layer (< 200 m)
(e.g. Kutsher et al., 2012), and thus GEM and GOM at night were continuously replenished by
those produced from daytime and remaining in the residual layer, which likely caused higher
nighttime values at PM. Daytime peaks of GOM at TF and AI were most likely caused by
photochemical oxidation of GEM under strong solar radiation. The causes for such variation were
examined in Sect. 3.4.2.
**3.2 Simulated diurnal variation and speciation of GOM**

Model simulated diurnal cycles of GOM averaged over the 50, 12, and 21 clear-sky days

at AI, TF, and PM, respectively, were shown in Fig. 2. The patterns of diurnal variation were
similar at the three sites with small discrepancy on the occurring time of daily peaks (~ 13:00 LT
at AI, and ~14:00 LT at TF and PM), but the magnitude varied large by site. AI had the largest
GOM diurnal amplitude (i.e., daily maximum – daily minimum) ranging from 0.73 to 13.29 ppqv,
TF from 0.05 to 0.57 ppqv, and PM showed a very small range from 0.05 to 0.14 ppqv. Similar
magnitude variation was also exhibited in GOM observations (Fig. 2). Overall, simulated GOM
mixing ratios at the three sites were in agreement with observations (detailed comparison in Sect.

3.3).

The simulations suggested that the dominant GOM species and GEM oxidants varied by

site (Fig. 3). At AI, brominated GOM species comprised 59–81% of the total GOM over 08:00–
18:00 EDT, whereas HgO was dominant (50–92% of the total GOM) during the remaining day.
At TF and PM, HgO was the predominant GOM species (62–88%). HgO was produced from
oxidation of GEM by $O_3$ and OH. The contribution to HgO from oxidation by $O_3$ was larger than
by OH except at noon when OH mixing ratios reach daily peaks resulting in comparable
contributions (48 and 52% by OH and $O_3$, respectively). At AI, $BrHgNO_2$ and HgBrO were the
most abundant brominated GOM species, which constituted ~ 96% of the total brominated GOM.
HgBrO was produced from the GEM + BrO reaction, while $BrHgNO_2$ were produced from GEM
oxidation by Br radicals followed by reactions of HgBr with $NO_2$. $Hg(OH)_2$ from GEM oxidation
by $H_2O_2$ appeared to be an important nighttime GOM species at the inland site (PM), accounting
for 33% of the total GOM at night. Other GOM species were negligible in the studied cases.
**3.3 Model evaluation**
For all cases at AI and TF, the average simulated and observed GOM diurnal cycles agreed
reasonably well in both magnitude and shape, whereas at PM the model appeared to have missed
both (Fig. 2). Three salient features were noted for the disagreement between the model and
observational results. First, the standard deviation of observed GOM mixing ratios was a factor of
2–7 larger than that of the simulated. This suggested that the model could capture the mean values
of GOM, but not the very low and very large mixing ratios. Second, observed nighttime GOM
mixing ratios were 12–200% larger than the simulated at AI, indicating that the model did not
capture certain nighttime processes producing GOM in the MBL. Third, the simulated diurnal
cycle was the opposite of the observed at PM, with the maximum during the day and minimum at
night. It was likely that the model simply simulated the dependence of GOM production on solar
radiation. At PM, more processes may have contributed to the diurnal variation. At night, the site
is above the nocturnal boundary layer and exposed to the GOM produced in the preceding
convective boundary layer, which could continually replenish surface GOM at the site that was
lost via dry deposition and perhaps reduction. The model-observation discrepancies of GOM at
the three sites were discussed as follows.
3.3.1 Appledore Island (marine)
Of the 50 cases at AI, 27 diurnal cycles of GOM were simulated with the average values
and patterns close to the observed and $NMSE_s = 1.88\%$, denoted as *matching* cases hereafter, 8
were underestimated with $NMSE_s = 121\%$, and 15 were overestimated with $NMSE_s = 171\%$. The
observed and simulated average GOM mixing ratios and the corresponding ranges were calculated
for the matching, under-estimation, and over-estimation cases at AI (Fig. 4a ). For more than half
of the time (27 matching cases out of 50 cases in total), the model captured the average GOM
diurnal cycle, the diurnal cycle pattern and overall GOM levels. Beyond that, Fig. 4a shows large
difference in the observed GOM levels among the matching, under-estimation, and over-
estimation cases. On average, the observed daytime peak in the under-estimation cases was about
twice as large as that for the matching cases and 7 times larger than that for the over-estimation
cases. However, such difference was not captured by the model, suggesting that some GOM
producing processes in the MBL were not included or not realistically represented in the box model.
In addition, the GOM diurnal pattern in the over-estimation cases was different from those in the
under-estimation and matching cases. The average observed GOM diurnal cycles of the under-
estimation and matching cases both exhibited a daily maximum at 13:00 EDT and a minimum over
04:00–08:00 EDT, whereas the over-estimation cases showed a daily maximum at around 20:00
EDT and a minimum at 07:00–08:00EDT.
Such differences were due possibly to the challenges of simulating Br and BrO in the MBL
at AI. No measurements of Br and BrO radicals as well as $Br_2$ were available at AI. To reasonably
simulate mixing ratios of Br and BrO, $Br_2$ mixing ratios were calculated based on the BrO
observations at a mid-latitude MBL site from Saiz-Lopez et al. (2006), which was ~5.6 ppqv during
the daytime (06:00–21:00 EDT). Saiz-Lopez et al. (2006) showed that daytime peak mixing ratios
of BrO in the MBL could vary by a factor of 2 over a time period of 3 days. Such variation was
not captured in our box model, potentially resulting in uncertainty of up to 100% in simulated Br
mixing ratios with subsequent effects on GOM simulation.
In the over-estimation cases, the simulated GOM daytime peaks were very low and
appeared later during the day than in the under-estimation and matching cases. Considering the
late afternoon peak (17:00 EDT) of $O_3$ compared to the noontime peak of Br radicals, $O_3$ possibly
played a more important role in the over-estimation cases. To verify this hypothesis, a sensitivity
simulation was conducted without the initial $Br_2$ mixing ratio fixed for these cases, termed as the
$O_3$/OH scenario. In this sensitivity runs, the $Br_2$ concentration rapidly diminished with time leading
to very low concentrations of Br and BrO. The $O_3$/OH scenario turned out to better represent these
15 overestimation cases with $NMSE_s$ = 34% (compared to 167% with $Br_2$ mixing ratio fixed).

These sensitivity simulations suggested that in the MBL, Br may be a dominant GEM

oxidation most of the time, but at times of low Br mixing ratios, $O_3$ could become dominant. To
identify the origin of the air masses at AI, backward trajectory analysis was conducted using the
HYSPLIT4 model (https://ready.arl.noaa.gov/HYSPLIT.php). All 24 h backward trajectories
started from the time of GOM daily peaks for the 50 cases. The trajectory results were clustered
for over-estimation, matching, and under-estimation cases (Fig. 5). Based on these trajectories, in
about half of the 15 over-estimation cases air masses originated from marine environments, while
in more than 80% of the 27 matching cases and 7 out of 8 under-estimation cases air masses came
from inland northwest of AI. Note that in those under-estimation cases GOM mixing ratios were
exceptionally large, exceeding 30 ppqv.

Different source areas of air masses reaching AI could be one of the reasons for the large

variation of GOM observations. The highest levels of GOM were observed in summer with RH
roughly < 50% at AI (Mao et al., 2012). A close examination of the 50 cases at AI revealed low
RH levels ($\leq$ 45%) on 16 days. The time periods with RH $\leq$ 45% appeared mostly (78% of the
time) in the afternoon over 12:00–20:00 EDT and less so (22%) at night over 21:00–02:00 EDT.
During these time periods, increased GOM (15 out of 16, compared with periods with high RH on
the same day) and daily maximum GOM (10 out of 16) occurred simultaneously at low RH,
regardless of the time of the day.

Interestingly, the RH level of 45% corresponds to the crystallization point of NaCl (Cziczo

et al., 1997; Tang et al., 1997). The crystallization of sea-salt aerosols might be link to the very
high GOM peaks in certain ways. Rutter and Schauer (2007) found that particles of potassium and
sodium chlorides had high partitioning coefficients that could shift the GOM gas-particle
partitioning toward the aqueous phase, while ammonium sulfate, levoglucosan, and adipic acid
would shift the partitioning toward the gas phase. It was thus *hypothesized* that, when these inland
air masses reached the MBL mixed with the marine air, the processes discussed above might have
been activated involving the interaction between land and marine air, which potentially resulted in
those very high GOM mixing ratios.

Laskin et al. (2012) found effective reactivity of chloride ($Cl^-$) components with organic

acid in sea salt aerosols (SSAs), possibly leading to depletion of $Cl^-$ and formation of organic salts
in aerosols. Biogenic compounds in air masses originating from inland forested areas could be
oxidized forming organic acids in transit. As inland air reached the MBL, these organic acids
would deposit onto SSAs and could subsequently change SSAs' chemical and physical properties,
such as lowering concentrations of $Cl^-$ and forming a thick organic film on the outside of SSAs.
The lower concentrations of $Cl^-$ and higher concentrations of organic acid in aerosols might have
contributed to the shift in the gas-to-particle partitioning to the gas phase and resulted in higher
GOM mixing ratios in the atmosphere.

Another possible explanation could be air masses of inland origin encountering marine air

rich in atmospheric Br and BrO radicals. The main source of atmospheric Br is thought to come
from the release of $Br_2$ and BrCl from SSA (Finlayson-Pitts, 2010; Sander et al., 2003).
Experimental studies suggested $Br^-$ enhancements of a factor of 40 to 140 on the surface of
sufficiently dry artificial SSA (Ghosal et al., 2008; Hess et al., 2007). Therefore, when drier inland
air masses were mixed with marine air in the MBL under relatively low RH conditions, SSA
became drier, forcing more $Br_2$ to be released from SSA, resulting in enhanced oxidation of GEM
by Br and BrO radicals. These hypotheses need to be validated in future research. These
mechanisms are presently missing in the box model, leading to the model's inability to capture
very high GOM mixing ratios. Measurements of halogen species and a better gas-particle
partitioning mechanism are needed to better the model's performance.
3.3.2    Thompson Farm (coastal)
Generally, the box model performed well at TF (Fig. 2) with overall $NMSE_s = 0.75\%$ and
RMSE= 0.78 ppqv. Of the 12 cases at TF, 6diurnal cycles of GOM (50%) were simulated
reasonably well with $NMSE_s <$ 50 %, 2 were underestimated by $\sim$ 70 %, and 4 cases were
overestimated by a factor of 2to 5. Overall, the observed average diurnal cycles of GOM for all
selected summer clear-sky days at TF had daily peaks during 14:00–20:00 EDT with very low
values at night between 0:00 and 8:00 EDT (Sigler et al., 2009) (Fig. 2). The peak observed at
17:00 EDT (Fig. 2) was largely affected by the abnormally high GOM peak in that one under-
estimation case (Fig. 4b).
For the over-estimation and matching cases, the model reproduced very low GOM mixing
ratios at night (Fig. 4b). For the same reason substantially lowering GEM mixing ratios at night
and in the early morning at TF (Mao et al., 2008), the low nighttime GOM at TF was probably
caused by loss via dry deposition under nocturnal inversion. To capture these low values in model
simulations, realistic nocturnal boundary layer height data were needed beside solid representation
of dry deposition and chemistry in the model. The diurnal cycle of boundary layer height in the
box model was parameterized based on reanalysis data obtained from the Research Data Archive
at the National Center for Atmospheric Research (http://rda.ucar.edu/datasets/ds093.0/). Use of
these data helped to reproduce the low nighttime GOM levels in simulations for the TF site.
Another notable feature in Fig. 4b is the exceedingly high observed GOM mixing ratios in the
under-estimation cases and the low observed GOM mixing ratios throughout the day in all over-
estimation cases. Observed GOM mixing ratios in the under-estimation cases showed a factor of
3–4 larger than those in the matching cases, and a factor of 3–31 larger than those in the over-
estimation cases (Fig. 4b). Concurrently, larger fine particle concentrations, 7468 $cm^{-3}$ on average,
were observed for the under-estimation cases, which was 51 and 80% larger than those in the
matching cases and over-estimation cases, respectively. Lower RH, 66% on average, was observed
in the under-estimation cases, 5and 11% lower than that in the matching and over-estimation cases,
respectively. Moreover, higher air pressure (1018, 8 and 12 hPa larger than the matching and over-
estimation cases, respectively), lower wind speed (0.8 m $s^{-1}$ on average, 35 and 68% lower than
matching and over-estimation cases respectively), and stronger solar radiation flux (8 and 13%
stronger than matching and over-estimation cases respectively) were found in the under-estimation
cases. An examination of the sea level pressure maps (Figure S1) in the under-estimation cases
suggested that these cases occurred under the strongest Bermuda High influence, with the calmest,
sunniest, and driest conditions of all cases, which is most conducive to photochemistry and
pollution build-up that may have ultimately contributed to the very large GOM mixing ratios in
those under-estimation cases. Our model appeared to fail to mimic the chemistry under such
conditions that produced the largest GOM mixing ratios.
3.3.3 Pack Monadnock (inland, rural, elevated site)
At PM, diurnal cycles of GOM were overestimated with $NMSE_s$ = 70% and overall
RMSE= 0.13 ppqv. However, considering the extremely low mixing ratios of GOM observed at
PM (Fig. 2), cases with RMSE < 0.1 ppqv (LOD) were considered as matching cases. Therefore,
the model reasonably simulated 11 out of 21 (52%) cases, underestimated in 1, and overestimated
in 9. Evaluation of simulated GOM diurnal cycles against observations (Fig. 2) showed reasonable
agreement with general overestimation ranging over 0.05–0.07 ppqv.

The observed GOM diurnal cycle (Fig. 2f) showed daily maximums at 08:00 and 23:00

EDT, which were mainly influenced by the underestimated case (Fig. 4c). In comparison, the
remaining (95 %) cases showed a very flat GOM diurnal cycle at PM. The first and most important
reason for such observation-model discrepancy is that the PM site is a mountain site (700 m a.s.l.),
which is above the nocturnal inversion layer (~200m at TF) but within the convective boundary
layer during the day. At night, a regional pool of GOM produced in the preceding convective
boundary layer remained in the residual layer, which kept the surface GOM levels from dropping
below the LOD at night at PM. The slight decline of GOM mixing ratios after sunrise was because
of mixing with the lower altitude air masses with depleted GOM from the night. The effect of the
PM's site characteristics was not represented in the box model, which could result in model's
inability to simulate diurnal variation associated with this aspect of the site. In addition, due to the
dominance of GEM oxidation by $O_3$ in GOM production in the model, it was highly likely that the
flat diurnal cycles (slightly higher at night) of GEM (Fig. 2) and $O_3$ were mirrored in GOM mixing
ratios.

**3.4   Sensitivity analysis**
3.4.1   Sensitivity of GOM to physical and chemical parameters

The base scenario (*Scenario* 0) of these sensitivity runs represented the real atmospheric

conditions on the selected 50 days at AI. *Scenarios* 1–10 are sensitivity runs where one parameter
in the base  scenario was changed at the time (Table 3). *Scenario* 1 turned off photolysis reactions.
*Scenarios* 2–4 tested the gas-particle partitioning scheme. The liquid water content range was
derived from Hedgecock et al. (2003). *Scenarios* 5–9tested the sensitivity of GOM mixing ratios
to GEM oxidation reactions and their coefficients. *Scenarios* 10–11 tested the sensitivity of GOM
mixing ratios to temperature. The temperature range was based on the observed average
temperature diurnal cycle.
The importance of photochemical radicals in GEM oxidation was demonstrated clearly in
decreases of 3-92 and 2-100 % in daytime GOM and PBM, respectively with largest decreases at
noon as a result of turning off photochemistry (*Scenario 1*). *Scenario 2* showed ~74% of oxidized
Hg transformed to PBM at AI with gas-particle partitioning switched on. In this scenario, HgO
and $Hg(OH)_2$ were more sensitive than halogenated GOM species (such as $BrHgNO_2$). Turning
off gas-particle partitioning more than quadrupled the mixing ratios of HgO and $Hg(OH)_2$
throughout the day compared to increases of more than 100 and 60% halogenated GOM species
during daytime and nighttime, respectively.
Decreasing liquid water content by 1 order of magnitude tripled GOM mixing ratios,
whereas increasing the same amount decreased GOM by 80% (*Scenarios 3–4*). Sensitivity of
GOM and PBM mixing ratios to dominant GEM oxidation reactions are shown in *Scenarios 5–9*.
Using the slowest rate coefficient of GEM + $O_3$ obtained from Hall (1995), as opposed to the one
from Snider et al. (2008) led to a decrease of 56.7% in HgO, and decreases of 15 and 85% in total
GOM during daytime and nighttime, respectively. Using an order of magnitude faster rate
coefficient of GEM + Br from Ariya et al. (2002) increased 250% of total GOM during daytime.
Turning off GEM oxidation by $O_3$, OH, or Br resulted in decreases of 16, 10, and 48%, respectively,
in daytime GOM mixing ratios. Turning off the GEM + Br oxidation reaction also decreased
daytime PBM mixing ratios by 60%. However, for nighttime GOM and PBM mixing ratios,
turning off the GEM + $O_3$ reaction caused decreases of 88 and 51%, respectively, since Br and OH
are both photochemical radicals and $O_3$ was the predominant oxidant for GEM in the model.
*Scenarios 10–11* suggested that nighttime GOM and PBM mixing ratios were more
sensitive to temperature than those during daytime. Increasing temperature by 10 K caused a 9%
increase each in GOM and PBM mixing ratios during daytime but a decrease of 13% in GOM and
54% in PBM at night. This was because the rate coefficient of GEM + $O_3$ increases with increasing
temperature, but the rate coefficient of GEM + OH decreases with increasing temperature.
In summary, the parameters used in gas-particle partitioning process, including solar
radiation values, temperature, and the rate coefficients of major GEM oxidation reaction, could all
affect the GOM simulation but with varying degree. Aerosol properties were suggested to play a
very important role in the partitioning of ambient GOM and PBM species and thus should be better
represented in future Hg model simulation studies. Using a slower rate coefficient of GEM + $O_3$
(Hall, 1995) had similar effects as not including the GEM + $O_3$ reaction, i.e. decreasing GOM
mixing ratios, especially at nighttime, and brominated GOM species becoming dominant. The
GEM + OH reaction was not as important as GEM + $O_3$ or Br. The use of a higher GEM + Br rate
coefficient derived from the study by Ariya et al. (2002) caused more than a factor of 3 higher
GOM and PBM resulting in overestimated GOM for most cases. GOM and PBM production
appeared to favor lower temperature at daytime and higher temperature at night, and simulated
GOM concentrations were not as sensitive to temperature change as to solar radiation and gas-
particle partitioning.
3.4.2   Influence of physical and chemical processes on GOM diurnal cycle
Large variations were exhibited in both observed and simulated GOM mixing ratios at AI,
TF, and PM (Fig. 2). Considering that all cases were under relatively calm, clear-sky conditions,
the simulated GOM mixing ratio and diurnal cycle were controlled primarily by chemical reactions,
dry deposition, and gas-particle partitioning. To quantify the contribution of processes to the
difference of GOM mixing ratios at the three sites, two sensitivity scenarios were conducted: use
the same physical parameters as those of AI for TF (denoted as TF_AIaerodry) and PM (denoted
as PM_AIaerodry).
Comparison of simulated GOM diurnal cycles from the AI, TF_AIaerodry and
PM_AIaerodry scenarios showed the influence of different chemical scenarios on GOM mixing
ratios at the three sites. At night, GOM mixing ratios at the three sites did not vary significantly
(0-2 ppqv), with higher values at PM than those at AI and TF (Fig. 6). However, the mid-day peak
at AI was more than a factor of two greater than those in the PM_AIaerodry and TF_AIaerodry
scenarios, indicating more chemical transformation of Hg occurring at AI. The daytime mixing
ratios of GOM at TF and PM were similar, while the nighttime GOM mixing ratios at PM were
30-52% higher than at AI and 20-200% higher than at TF. This probably resulted from larger
nighttime GEM and $O_3$ mixing ratios, hence producing more GOM, at PM than at TF and AI.
Specifically, nighttime GEM mixing ratios at PM were 8-15% higher than at AI and 8-34% higher
than TF cases, while nighttime $O_3$ mixing ratios at PM were 11-70% larger than at AI and 35-260%
larger than at TF. PM had higher nighttime GEM and $O_3$ mixing ratios, because this site was
exposed in the residual boundary layer at night due to its high elevation, constantly replenished
with the regional pool of air from the preceding convective boundary layer. Overall, chemical
transformation contributed ~60% of the daytime difference in GOM between AI and the two sites
over land (TF and PM), 33% of the nighttime difference between AI and TF, and 26% of the
difference between PM and AI.
In summary, the sensitivity scenarios suggested that dry deposition and gas-particle
partitioning contributed 4-37% and 30-96%, respectively, of the total GOM difference between AI
and PM. Both processes had larger contributions at night that during daytime. Dry deposition
contributed 6-24% of the GOM difference between AI and TF and gas-particle partitioning 18-

78%.

3.4.3   Br chemistry in the MBL
Diurnal cycles of Br and BrO radicals (Fig. 7) were simulated using the Br chemical
mechanism described in Sect. 2. Photodissociation of $Br_2$ was the main source of Br and BrO
radicals during daytime. Our simulations suggested that reactive Br compounds were significant
gaseous oxidants of GEM in the MBL at a fixed initial mixing ratio of 5.6 ppqv for $Br_2$. Increasing
initial mixing ratios of $Br_2$ by 25% resulted in an increase of 0.01–2.15 ppqv in GOM mixing ratios.
In addition, the reaction of BrO with methyldioxy ($CH_3O_2$) radicals could have important
influence on the mixing ratios of Br, BrO, and GOM. Simulated daytime mixing ratios of $CH_3O_2$
was ~ 40 pptv, and the rate coefficient of $(5.7\pm0.6) \times 10^{-12}$ cm$^3$ molecule$^{-1}$ s$^{-1}$ at 298 K for BrO +
$CH_3O_2$ (Aranda et al., 1997) was used for our simulations. Pathways B1, B2, and B3 were
suggested by Aranda et al. (1997) based on an experimental study (Table 4). However, the
production of $CH_3O$ may be due to its self-reaction in B1. Guha and Francisco (2003) suggested
$CH_3OOOBr$ to be a likely intermediate of this reaction, and that $CH_3OOOBr$ could dissociate to
$CH_2O$+HOOBr (B4, Table 4). Based on thermodynamics calculations, $CH_3OBr$ and $O_2$ (B3, Table
4) were possible products. BrOO and HOBr were both included in the Br chemical cycle and can
be transformed back to Br and BrO radicals in the model. However, it is unclear whether $CH_3OBr$
(product of B3) or HOOBr (product of B4) could be transformed back to Br and BrO radicals in
the atmosphere. In this case, using the B3 or B4 pathway did not appear to make a difference in
our box model results.
In this study, the B1 and B2 pathways were used for the $CH_3O_2$ +BrO reaction as part of
the base scenario (denoted as Sim-avg BrOO). The sensitivity run Sim-avg $CH_3OBr$ used the B3
pathway in lieu of B1 and B2. The simulated average and the range of GOM diurnal cycles in the
base and sensitivity scenarios were evaluated against observed mean and median GOM diurnal
cycles of the 50 study cases at AI (Fig. 8). If the $CH_3O_2$ +BrO reaction followed the B1 and B2
pathways, this reaction had a negligible effect on reactive Br radicals. However, if B3 or B4 was
applied, the simulated total GOM mixing ratio was lowered by 50% during daytime. Moreover,
the simulated GOM diurnal cycle in the base scenario agreed favorably with the observed average
GOM diurnal cycle (NMSE= 15 %), while the results of the Sim-avg $CH_3OBr$ scenario were in
better agreement with the observed median GOM diurnal cycle (NMSE= 14%). These agreements
indicated that, if the BrO+$CH_3O_2$ reaction was a net sink of BrO radicals, the model was able to
simulate most cases better, whereas if the product of BrO+$CH_3O_2$ was transformed back to Br or
BrO radicals, the model appeared to capture those cases with large GOM mixing ratios (> 6 ppqv).
Due to the scarcity of kinetic research on the B3 and B4 pathways, we used B1 and B2 pathways
for $CH_3O_2$ +BrO reaction in this study.
In short, the pathways of BrO+$CH_3O_2$ could play an important role in atmospheric Br
chemistry and Hg speciation in Br-rich environments. Research on the reaction pathways and rate
coefficients of the BrO+$CH_3O_2$ reaction is warranted to better assess the role of this reaction.
**4  Summary**
This study provided a state-of-the-art chemical mechanism with most up-to-date Hg and
halogen chemistry and tested the mechanism for three different environments using a mercury box
model. Eighty-three summer clear-sky days were selected at marine, coastal, and inland elevated
sites in southern New Hampshire to evaluate the model. As a result, for each of the three
environments, GOM diurnal cycles of over half selected cases were reasonably represented by the
box model. It was hypothesized, based on the key results and discussion presented in Section 3,
that dry air masses with organic compounds transported from inland may result in very large GOM
mixing ratios in the MBL possibly due to changing physical and chemical properties of sea salt
aerosols. The low nighttime and morning GOM mixing ratios at coastal site were likely a result of
a net loss due to dry deposition in the nocturnal inversion layer. The GOM mixing ratios above the
LOD at the inland site at night were probably caused by constant replenishment from a regional
pool, in the residual boundary layer, of GOM that was produced in the preceding daytime
convective boundary layer. The updated chemical mechanism largely improved the simulation of
the magnitude and pattern of GOM diurnal variation at the coastal and inland sites. HgO produced
from oxidation of GEM by $O_3$ and OH dominated GOM species at the coastal and inland sites,
while bromine-induced mercury species (mainly BrHgOOH, BrHgOBr, and HgBrO) were
important at the marine site. In Br chemistry, the products of the $CH_3O_2$ +BrO reaction strongly
influenced the simulated Br and Hg concentrations. In this study, GEM oxidation by $O_3$ and OH
was represented in ways similar to those in regional and global models, which is limited by the
current nebulous understanding of potential surface chemistry.

It should be noted that without measurements of speciated GOM, modeling results cannot

be used to conclusively identify the dominant oxidants of Hg, as well as dominant GOM species
in that matter, in the atmosphere. Indeed, the potential uncertainty in ambient Hg measurements
especially GOM is a major concern in the community. That being said, it is unlikely to have a
quantitative understanding of the bias of our GOM concentrations. Recent laboratory experiments
and reviews (Lyman et al., 2010; Jaffe et al., 2014; McClure et al., 2014; Huang and Gustin, 2015;
Gustin et al., 2015) reported $O_3$ and relative humidity (RH) interferences on mercury halides for
KCl-coated denuder, the part of Tekran 1130 unit commonly used for GOM field measurements.
As stated in Section 2, in our GOM measurement the RH effect was minimized by adding
refrigeration to remove excess of water in the airsteam. $O_3$ interference and bias low GOM
collection efficiency of KCl-coated denuders were limited to a handful of GOM species in
laboratory experiments and remain untested in field measurements. If the measured GOM
concentrations were indeed biased low by a factor of 2 or 3 under certain conditions as previous
studies speculated, the matching cases at AI and TF would be reduced from 50% of the total cases
to 30%, and the model would potentially underestimate GOM concentrations in the remaining
cases (70%) by a factor of 3 to 4. It is however hard to speculate the effect at PM since most GOM
observations there were below the LOD. This suggested even greater unknowns in our
understanding of Hg chemistry. Therefore, more experimental or theoretical studies on Hg
reactions and better GOM measurement data are warranted to improve our understanding and
subsequently model simulations of atmospheric Hg cycling, which can ultimately serve policy-
making in an effective manner.
**Acknowledgements**
This work is funded by NSF AGS grant # 1141713. We thank T. Dibble, Y. Zhou, Y. Zhang, and
C. B. Hall for valuable suggestions and help. We are grateful to the three anonymous reviewers
for their thoughtful, detailed, constructive comments, which helped to improve the clarity of the
paper.

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

Table 1. Gas phase Hg reactions in the box model

| No. | REACTIONS | KINETIC ($cm^3 molecule^{-1}s^{-1}$) | REFERENCE |
|---|---|---|---|
| G1 | Hg $+ O_3 \rightarrow$ HgO $+ O_2$ | $8.43 \times 10^{-17} e^{-1407/T}$ | Snider et al., 2008 |
| G2 | Hg $+$ OH $\{+O_2\} \rightarrow$ HgO $+ HO_2$ | $3.55 \times 10^{-14} e^{294/T}$ | Pal and Ariya, 2004 |
| G3 | Hg $+ H_2O_2 \rightarrow$ Hg(OH)$_2$ | $8.5 \times 10^{-19}$ | Tokos et al., 1998 |
| G4 | Hg $+$ Cl $\rightarrow$ HgCl | $6.4 \times 10^{-13} e^{(680 \times (1/T - 1/298))}$ | Donohoue et al., 2005 |
| G5 | Hg $+ Cl_2 \rightarrow HgCl_2$ | $2.6 \times 10^{-18}$ | Ariya et al., 2002 |
| G6 | Hg $+$ Br $\rightarrow$ HgBr | $3.7 \times 10^{-13} (T/298)^{-2.76}$ | Goodsite et al., 2004, 2012 |
| G7 | Hg $+$ BrO $\rightarrow$ HgBrO | $1.8 \times 10^{-14}$ | Raofie and Ariya, 2004 |
| G8 | Hg $+$ I $\rightarrow$ HgI | $4.0 \times 10^{-13} (T/298)^{-2.38}$ | Goodsite et al., 2004 |
| G9 | HgI $\rightarrow$ Hg $+$ I | $3.0 \times 10^{9} e^{-3742/T}$ | Goodsite et al., 2004 |
| G10 | HgBr $\rightarrow$ Hg $+$ Br | $1.6 \times 10^{-9} e^{-7801/T} \times [M]$ | Dibble et al., 2012 |
| G11 | HgBr $+$ Br $\rightarrow$ Hg $+ Br_2$ | $3.89 \times 10^{-11}$ | Balabanov et al., 2005 |
| G12 | HgBr $+$ Br $\rightarrow$ HgBr $+$ Br | $3.97 \times 10^{-11}$ | Balabanov et al., 2005 |
| G13 | HgBr $+$ Br $\rightarrow$ HgBr$_2$ | $2.98 \times 10^{-11}$ | Balabanov et al., 2005 |
| G14 | ClO $+$ HgCl $\rightarrow$ ClHgOCl | $5.0 \times 10^{-11}$ | Dibble et al., 2012 |
| G15 | ClO $+$ HgBr $\rightarrow$ BrHgOCl | $5.0 \times 10^{-11}$ | Dibble et al., 2012 |
| G16 | BrO $+$ HgCl $\rightarrow$ BrHgOCl | $1.09 \times 10^{-10}$ | Dibble et al., 2012[1] |
| G17 | BrO $+$ HgBr $\rightarrow$ BrHgOBr | $1.09 \times 10^{-10}$ | Dibble et al., 2012; Wang et al., 2014 |
| G18 | NO$_2$ $+$ HgCl $\rightarrow$ ClHgNO$_2$ | $8.6 \times 10^{-11}$ | Dibble et al., 2012[1] |
| G19 | NO$_2$ $+$ HgBr $\rightarrow$ BrHgNO$_2$ | $8.6 \times 10^{-11}$ | Dibble et al., 2012; Wang et al., 2014 |
| G20 | HO$_2$ $+$ HgCl $\rightarrow$ ClHgOOH | $8.2 \times 10^{-11}$ | Dibble et al., 2012[1] |
| G21 | HO$_2$ $+$ HgBr $\rightarrow$ BrHgOOH | $8.2 \times 10^{-11}$ | Dibble et al., 2012; Wang et al., 2014 |
| G22 | OH $+$ HgCl $\rightarrow$ ClHgOH | $6.33 \times 10^{-11}$ | Dibble et al., 2012[1] |
| G23 | OH $+$ HgBr $\rightarrow$ BrHgOH | $6.33 \times 10^{-11}$ | Dibble et al., 2012; Wang et al., 2014 |
| G24 | IO $+$ HgBr $\rightarrow$ BrHgOI | $4.9 \times 10^{-11}$ | Wang et al., 2014 |

[1] The kinetic data of these HgCl reactions were not included in Dibble et al., 2012, they were assumed as the same
kinetic as the HgBr reactions, which were calculated by Wang et al. (2014).

Table 2. Box model input and simulated variables.

| Parameter | Appledore Island (AI) | Thompson Farm (TF) | Pack Monadnock (PM) |
|---|---|---|---|
| Observed[1] | | | |
| RH, relative humidity | 76.9±5.4 | 69.9±19.5 | 69.0±13.1 |
| Temperature, °C | 19.1±1.7 | 21.3±4.3 | 18.5±3.3 |
| [GEM], ppqv | 133.9±3.3 | 138.4±12.8 | 149.6±3.2 |
| [$O_3$], ppbv | 37.4±8.8 | 32.7±15.7 | 45.0±4.2 |
| [NO], pptv | 154.5[2] | 232.4±364.1 | 85.3±35.8 |
| [CO], ppbv | 169.6±13.9 | 156.2±10.8 | 120.2±7.2 |
| Simulated[3] | | | |
| [Br], ppqv | 28.50 | 0.20 | 0.18 |
| [OH], ppqv | 100.7 | 75.8 | 73.5 |
| Other[4] | | | |
| $v_d$, cm s$^{-1}$, dry deposition velocity | GEM – 0.0045 | GEM – 0.07 | GEM – 0.08 |
| | GOM – 0.5 | GOM – 1.2 | GOM – 2.0 |
| | PBM – 0.5 | PBM – 0.15 | PBM – 0.25 |
| H, M atm$^{-1}$, Henry's constants | HgO $-$ $3.2 \times 10^9$ | HgO $-$ $3.2 \times 10^9$ | HgO $-$ $3.2 \times 10^9$ |
| | $Hg(OH)_2$ $-$ $1.2 \times 10^7$ | $Hg(OH)_2$ $-$ $1.2 \times 10^7$ | $Hg(OH)_2$ $-$ $1.2 \times 10^7$ |
| | Other GOM $-$ $2.7 \times 10^9$ | Other GOM $-$ $2.7 \times 10^9$ | Other GOM $-$ $2.7 \times 10^9$ |
| L, $m_{water}^3 m_{air}^{-3}$, liquid water content | $5 \times 10^{-11}$ | $2.0 \times 10^{-11}$ | $1.25 \times 10^{-11}$ |
| $D_g$, m$^2$s$^{-1}$, diffusion coefficient | $1 \times 10^{-5}$ | $1 \times 10^{-5}$ | $1 \times 10^{-5}$ |
| Z, m, boundary layer height | 500 | $200 - 1120$[5] | 100 |
| $r_{dry}$, µm, dry aerosol radius | 3.5 | 0.3 | 0.07 |

[1] Observed 24-h mean values for all studied cases at these sites.
[2] Missing NO measurements at AI, use 154.5 ppqv for initial values.
[3] Simulated 24-h mean values for all studied days at these sites.
[4] Reference: Baumgardner et al., 2000; Kim et al., 2012; Mao and Talbot, 2004; Moldanová and Ljungström, 2001;
Pillai and Moorthy, 2001; Shon et al., 2005; Zhang et al., 2009, 2012.
[5] TF boundary layer height changed at each hour, the averaged diurnal cycle was obtained from Research Data Archive
at the National Center for Atmospheric Research, http://rda.ucar.edu/.


Table 3. Sensitivity scenarios with varying physical and chemical parameters. The superscript D represents daytime and N nighttime. Downward arrows stand for decreases and upward arrows increases. T stands for the temperature diurnal cycle in the base scenario, and T+10K or T-10K represents 10K higher temperature or 10K lower temperature throughout the day respectively.

| Scenario No. | photolysis | Gas-droplet partitioning | | Rate Coefficients ($cm^3$ $molec^{-1}$ $s^{-1}$) | | | Temp. | Results | |
| | | Include | Liquid water content ($m^3_{water} m^{-3}_{air}$) | GEM+$O_3$ (298K) | GEM+OH (298K) | GEM+Br (298K) | | GOM | PBM |
|---|---|---|---|---|---|---|---|---|---|
| **Base Scenario** | | | | | | | | | |
| 0 | Yes | Yes | 5.0E-11 | 7.5E-19[1] | 9.5E-14[2] | 3.7E-13[3] | T | -- | -- |
| **Photochemistry** | | | | | | | | | |
| 1 | No | Yes | 5.0E-11 | 7.5E-19 | 9.5E-14 | 3.7E-13 | T | ↓ 3%-92% [D] | ↓ 2-100% [D] |
| **Gas-particle partitioning** | | | | | | | | | |
| 2 | Yes | No | -- | 7.5E-19 | 9.5E-14 | 3.7E-13 | T | ↑ ~280% | ↓ 100% |
| **Liquid water content** | | | | | | | | | |
| 3 | Yes | Yes | 3.0E-12 | 7.5E-19 | 9.5E-14 | 3.7E-13 | T | ↑ ~200% | ↓ 80% |
| 4 | Yes | Yes | 3.0E-10 | 7.5E-19 | 9.5E-14 | 3.7E-13 | T | ↓ 80% | ↑ 50% |
| **Reactions** | | | | | | | | | |
| 5 | Yes | Yes | 5.0E-11 | 3.0E-20[4] | 9.5E-14 | 3.7E-13 | T | ↓ 15% [D] ↓ 85% [N] | ↓ 49% [N] |
| 6 | Yes | Yes | 5.0E-11 | -- | 9.5E-14 | 3.7E-13 | T | ↓ 16% [D] ↓ 88% [N] | ↓ 51% [N] |
| 7 | Yes | Yes | 5.0E-11 | 7.5E-19 | -- | 3.7E-13 | T | ↓ 10% [D] | Negligible |
| 8 | Yes | Yes | 5.0E-11 | 7.5E-19 | 9.5E-14 | -- | T | ↓ 48% [D] | ↓ 60% [D] |
| 9 | Yes | Yes | 5.0E-11 | 7.5E-19 | 9.5E-14 | 3.2E-12[5] | T | ↑ 250% [D] | ↑ 300% |
| **Temperature** | | | | | | | | | |
| 10 | Yes | Yes | 5.0E-11 | 7.5E-19 | 9.5E-14 | 3.7E-13 | T+10K | ↓ 9% [D] ↑ 13% [N] | ↓ 9% [D] ↑ 54% [N] |
| 11 | Yes | Yes | 5.0E-11 | 7.5E-19 | 9.5E-14 | 3.7E-13 | T-10K | ↑ 9% [D] ↓ 11% [N] | ↑ 8% [D] ↓ 28% [N] |

[1] Snider et al., 2008; [2] Pal and Ariya, 2004; [3] Goodsite et al., 2004, 2012; [4] Hall, 1995; [5] Ariya et al., 2002.

Table 4. Possible pathways of BrO + CH$_3$O$_2$ reaction

| NO. | Reactions | Kinetics (cm$^3$molecule$^{-1}$s$^{-1}$) | Reference |
|---|---|---|---|
| B1 | $BrO(g) + CH_3O_2(g) \rightarrow CH_3O(g) + BrOO(g)$ <br> $BrOO(g) \rightarrow Br(g) + O_2(g)$ | $1.4 \times 10^{-12}$ <br> Fast | Aranda et al., 1997; Atkinson et al., 2008 |
| B2 | $BrO(g) + CH_3O_2(g) \rightarrow CH_2O_2(g) + HOBr(g)$ | $4.3 \times 10^{-12}$ | Aranda et al., 1997; Atkinson et al., 2008 |
| B3 | $BrO(g) + CH_3O_2(g) \rightarrow CH_3OBr(g) + O_2(g)$ | ? | Aranda et al., 1997 |
| B4 | $BrO(g) + CH_3O_2(g) \rightarrow CH_3OOOBr(g) \rightarrow CH_2O(g) + HOOBr(g)$ | ? | Guha and Francisco, 2003 |

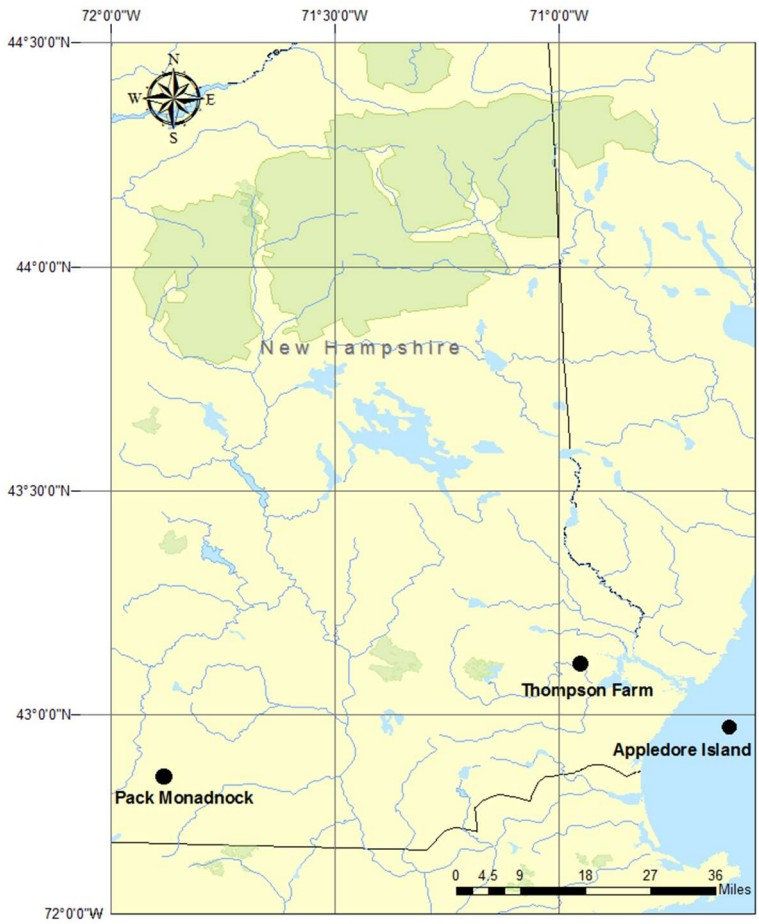

Figure 1. New Hampshire site map: Appledore Island (marine), Thompson Farm (coastal), and Pack Monadnock (inland elevated).

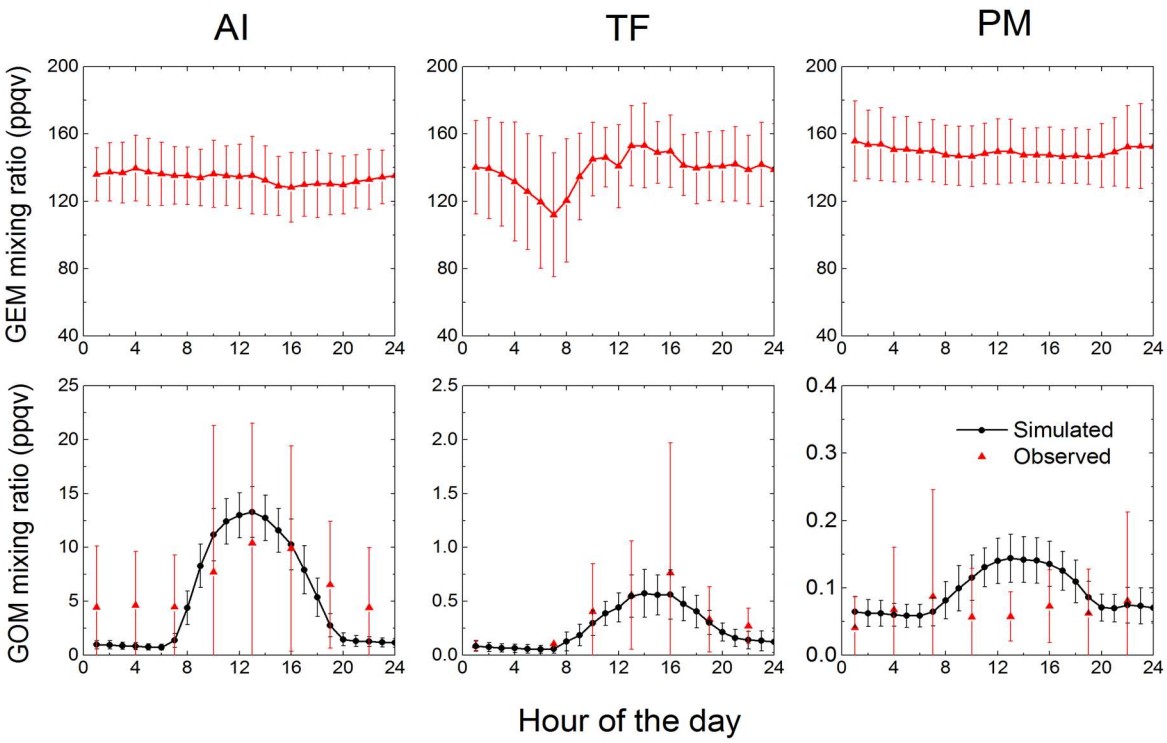

Figure 2. Average diurnal cycles of observed GEM (top panel) and simulated and observed GOM (bottom panel) averaged over the selected 50 days at Appledore Island (AI), 12 days at Thompson Farm (TF), and 21 days at Pack Monadnock ( PM) from summers of 2007, 2008, and 2010. The error bars represent standard deviation.

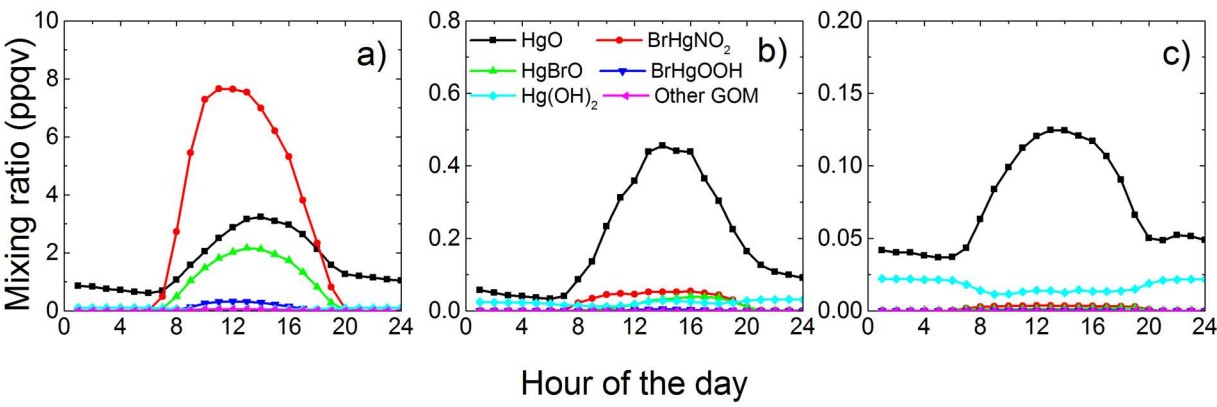

Figure 3. Simulated average diurnal cycles of GOM speciation at AI (a), TF (b), and PM (c).

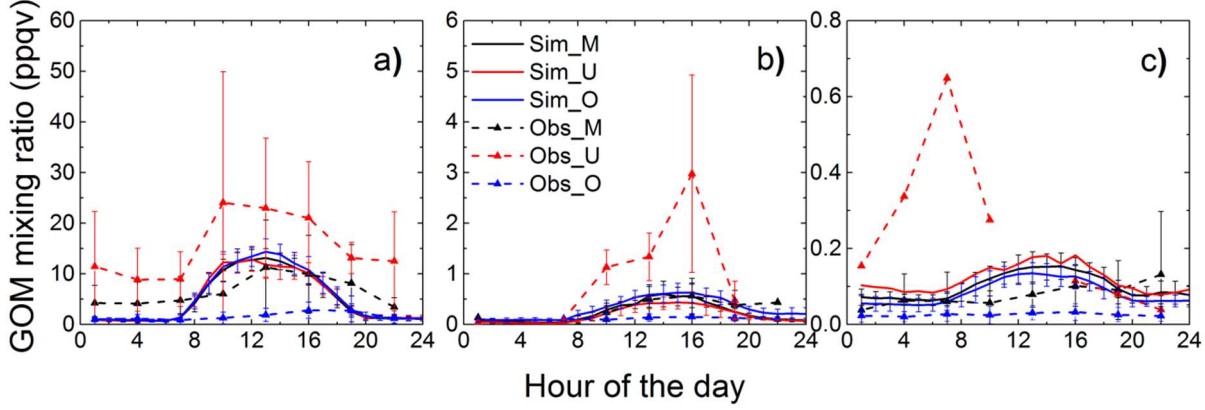

Figure 4. Observed (dash line with scatters) and simulated (solid line) average diurnal cycles of GOM for the matching (black, "Sim_M" and "Obs_M"), under-estimation (red, "Sim_U" and "Obs_U"), and over-estimation cases (blue, "Sim_O" and "Obs_O") at AI (a), TF (b), and PM (c). The bars represent the standard deviations at each hour for those specific days.

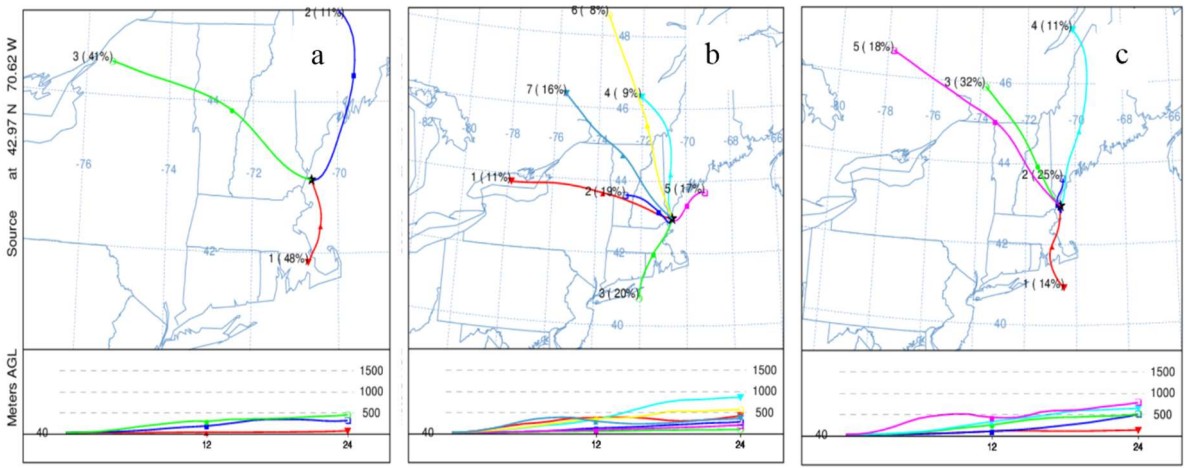

Figure 5.  Clustered 24-hours back trajectories of air masses in (a) over-estimation cases, (b) matching cases, and (c) under-estimation cases at AI.

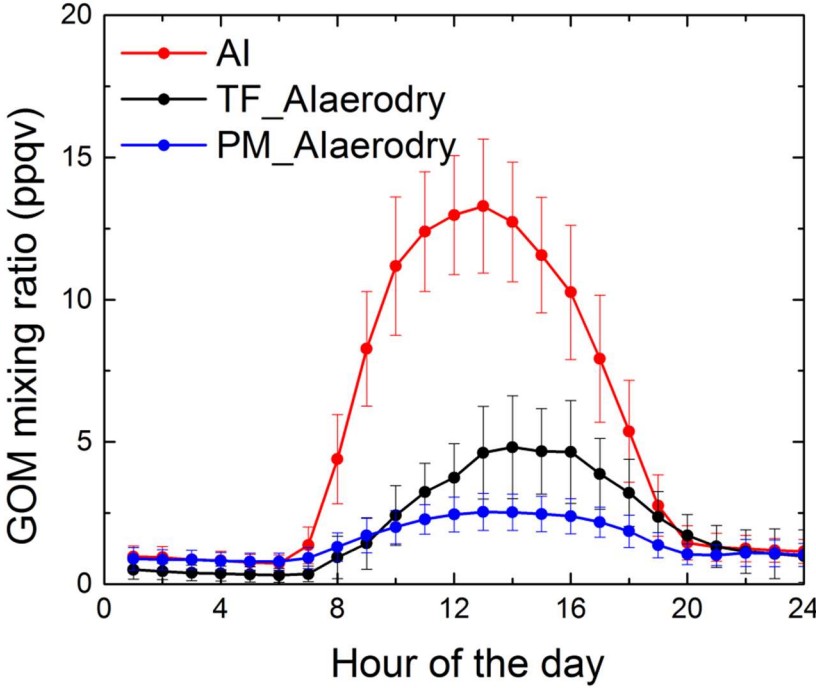

Figure 6. Simulated averaged diurnal cycles of GOM at AI (red), at TF (black) using AI dry deposition and gas-aerosol partitioning parameters ("TF_AIaerodry"), and at PM (blue) using AI dry deposition and gas-to-particle partitioning parameters ("PM_AIaerodry").

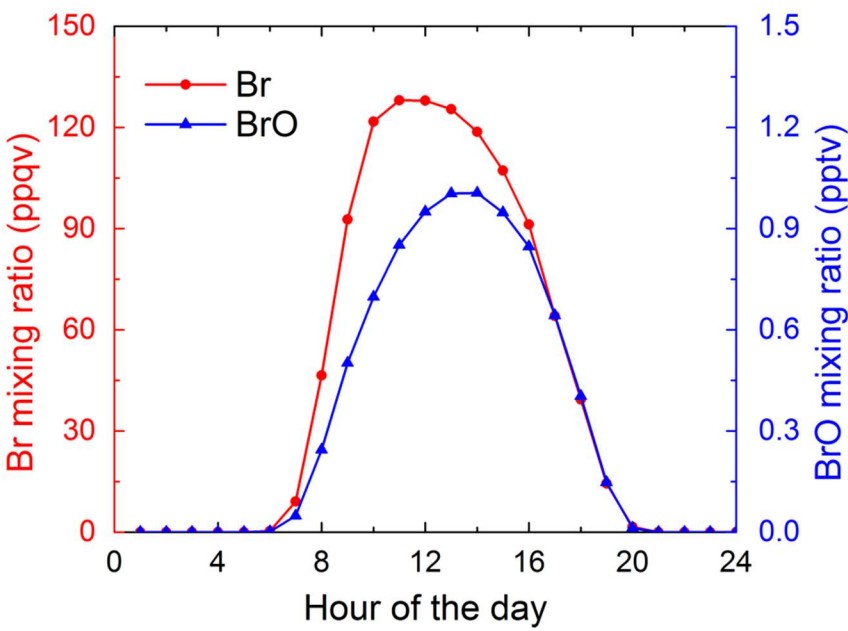

Figure 7.  Simulated diurnal cycles of Br (red) and BrO (blue) of the base case.

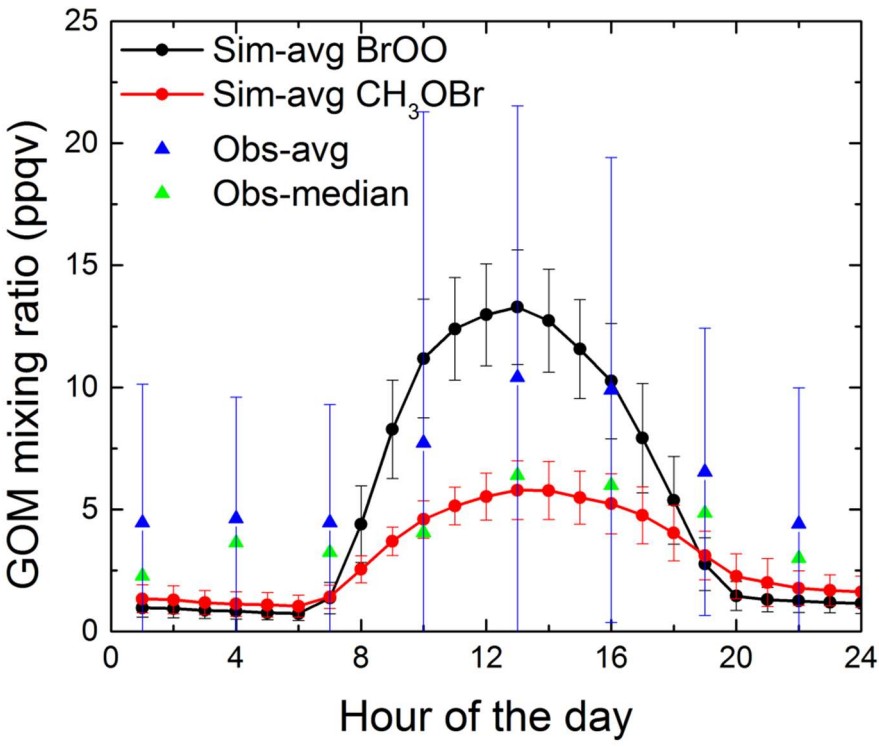

Figure8. Simulated average diurnal cycles of GOM for the base scenario ("Sim-avg BrOO", black circle) and for the "CH₃OBr" scenario ("Sim-avg CH₃OBr", red, circle), observed average GOM diurnal cycle ("Obs-avg", blue, triangle scatter), and observed median GOM diurnal cycle ("Obs-median", green, triangle scatter) of the 50 cases at AI.

Supplement:

Table S1. Aqueous phase reactions and equilibriums of Hg in the box model.

| No. | REACTIONS | KINETIC (L mol$^{-1}$ s$^{-1}$) or EQUILIBRIUM CONSTANT | REFERENCE |
|---|---|---|---|
| A1 | $Hg^0$ (aq) $+ O_3$ (aq) $\rightarrow$ HgO (aq) $+ O_2$ (aq) | $2.4 \times 10^9$ | Munthe et al., 1992 |
| A2 | $Hg^0$ (aq) $+$ OH (aq) $\rightarrow$ HgOH (aq) | $2.4 \times 10^9$ | Gardfeldt et al., 2001 |
| A3 | HgOH (aq) $+$ OH (aq) $\rightarrow$ $Hg(OH)_2$ (aq) | $1.0 \times 10^{10}$ | Nazhat and Asmus, 1973 |
| A4 | HgOH (aq) $+ O_2$ (aq) $+$ $H_2O$ (aq) $\rightarrow$ $Hg(OH)_2$ (aq) $+ H^+ + O_2^-$ | $1.0 \times 10^9$ | Nazhat and Asmus, 1973 |
| A5 | $Hg^0$ (aq) $+$ OH (aq) $\rightarrow$ $Hg^+ + OH^-$ | $2.0 \times 10^9$ | Lin and Pehkonen, 1997 |
| A6 | HgO (aq) $+ H^+ \rightarrow Hg^{2+} + OH^-$ | $1.0 \times 10^{10}$ | Pleijel and Munthe, 1995 |
| A7 | HOCl (aq) $+ Hg^0$ (aq) $\rightarrow Hg^{2+} + Cl^- + OH^-$ | $2.09 \times 10^6$ | Lin and Pehkonen, 1997 |
| A8 | $ClO^- + Hg^0$ (aq) $\rightarrow Hg^{2+} + Cl^- + OH^-$ | $1.99 \times 10^6$ | Lin and Pehkonen, 1997 |
| A9 | $HgSO_3$ (aq) $\rightarrow Hg^0$ (aq) $+$ S(VI) | 0.6 | Munthe et al., 1991 |
| A10 | $Hg(OH)_2$ (aq) $\rightarrow Hg^0$ (aq) $+$ products | $3.0 \times 10^7$ | Pleijel and Munthe, 1995 |
| A11 | $Hg^+ + HO_2$ (aq) $\rightarrow Hg^0$ (aq) $+ O_2$ (aq) $+ H^+$ | $1.0 \times 10^{10}$ | Xie et al., 2008 |
| A12 | $Hg^{2+} + HO_2$ (aq) $\rightarrow Hg^+ + O_2$ (aq) $+ H^+$ | $1.7 \times 10^4$ | Pehkonen and Lin, 1998 |
| AE1 | $Hg^{2+} + SO_3^{2-} \leftrightarrow HgSO_3$ (aq) | $2.0 \times 10^{13}$ | van Loon et al., 2001 |
| AE2 | $HgSO_3$ (aq) $+ SO_3^{2-} \leftrightarrow Hg(SO_3)_2^{2-}$ | $1.0 \times 10^{10}$ | van Loon et al., 2001 |
| AE3 | $Hg^{2+} + OH^- \leftrightarrow HgOH^+$ | $3.98 \times 10^{10}$ | Smith and Martell, 2004 |
| AE4 | $HgOH^+ + OH^- \leftrightarrow Hg(OH)_2$ (aq) | $1.58 \times 10^{11}$ | Smith and Martell, 2004 |
| AE5 | $HgOH^+ + Cl^- \leftrightarrow HgOHCl$ (aq) | $2.7 \times 10^7$ | Xiao, 1994 |
| AE6 | $Hg^{2+} + Cl^- \leftrightarrow HgCl^+$ | $2.0 \times 10^7$ | Smith and Martell, 2004 |
| AE7 | $HgCl^+ + Cl^- \leftrightarrow HgCl_2$ (aq) | $5.0 \times 10^6$ | Smith and Martell, 2004 |
| AE8 | $HgCl_2$ (aq) $+ Cl^- \leftrightarrow HgCl_3^-$ | 6.7 | Smith and Martell, 2004 |
| AE9 | $HgCl_3^- + Cl^- \leftrightarrow HgCl_4^{2-}$ | 13.0 | Smith and Martell, 2004 |
| AE10 | $Hg^{2+} + Br^- \leftrightarrow HgBr^+$ | $1.10 \times 10^9$ | Smith and Martell, 2004 |
| AE11 | $HgBr^+ + Br^- \leftrightarrow HgBr_2$ (aq) | $2.50 \times 10^8$ | Smith and Martell, 2004 |
| AE12 | $HgBr_2$ (aq) $+ Br^- \leftrightarrow HgBr_3^-$ | $1.50 \times 10^2$ | Smith and Martell, 2004 |

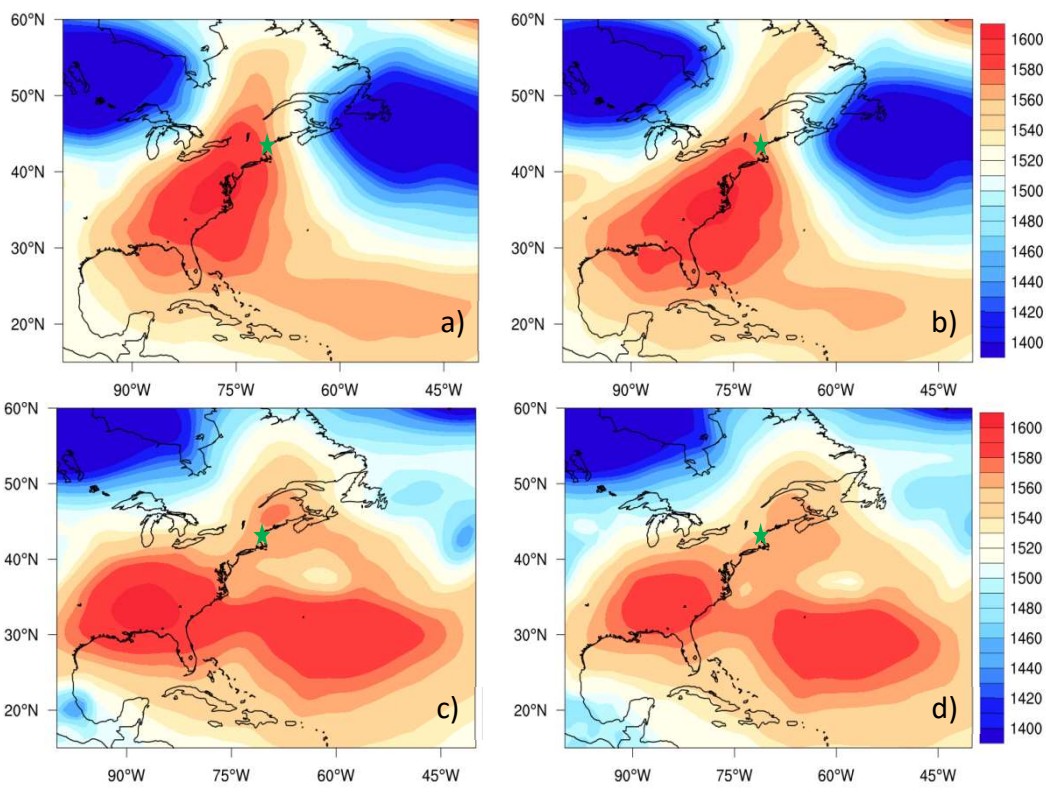

Figure S1. Geopotential height for a) 06/13/2008 08:00 EDT, b) 06/13/2008 14:00 EDT, c) 08/22/2007 14:00 EDT, and d) 08/22/2007 20:00 EDT at 850 hPa, the green star shows the location of TF site.