# Peer review of "Investigation of processes controlling summertime gaseous elemental mercury"

_Atmospheric Chemistry and Physics, 2015_

## Referee Comment (RC1) · Anonymous Referee #1 · 14 Feb 2016

This paper describes a model-measurement comparison study focusing on gaseous oxidized Hg (GOM) in the atmosphere at 3 sites in New Hampshire (marine, coastal, and inland). The main motivation for this study is to use updated reaction mechanisms and physical processes that control concentrations of GOM in the atmosphere and see if the model can reasonably reproduce the observations. There are many gaps in the understanding of what controls GOM in the atmosphere and these gaps are important to close since this species is readily wet and dry deposited and contributes to the burden of Hg accumulation in aquatic and terrestrial ecosystems. The photochemical Hg box model used in this work represents the state of the art with many updated reaction mechanism and physical processes. The sensitivity testing of the model output to

changing chemical and physical parameters is very good. The model reproduces the observations reasonably well, the most notable agreement is that the differing overall GOM measured concentrations between sites (AI > TF > PM) is reproduced in the model. The model also gives some indication of which chemical species comprise GOM at the different sites. This is new and valuable information if it can be corroborated by measurements. The main issue overall I have with the paper is that there is little discussion of the GOM measurements made at the 3 sites. At the end of the paper the authors acknowledge the latest papers suggesting that GOM may be systematically underestimated by the Tekran methods, but in the paper the authors do not indicate if there are any potential measurement biases in the data from the 3 sites, and if so how these might change the conclusions reached from the model-measurement comparison. This is a fundamental weakness of such a study that uses measurements with a high degree of uncertainty to validate (or compare to) a model that is also uncertain. That limitation aside, however, there is a great deal of value in such a comparison, and I find this paper to be largely acceptable as-is. Abstract: I understand word count in limited, but there should be some indication of what is new or novel about the research. Which finding contributes to our understanding the best? It should be more than a list of observations, but rather some indication of why these observations matter. Line 11, Page 3, provide reference(s) for this statement. Line 20 Page 6, "The ClO /BrO / IO radical cycles involve oxidation of Cl /Br / I radicals, photodissociation of ClNO2 / ClONO2 /BrNO2 /BrONO2 , production from other halogen radicals, and sink reactions to reproduce Cl/Br/O radicals or other halides." What is meant by "reproduce"? Do you mean to reproduce the observations? What if those observations are very uncertain? Line 15 Page 10, and Figure 2, There needs to be some discussion about the Tekran measurements. There were no mention of these in the methods section. Were the 3 instruments at the 3 locations operated in a consistent manner? Why such a large variation in GOM at AI, but very low levels at PM? The authors state the MDL for GOM is 0.1 ppqv yet most of the PM data in Figure 2 is < MDL. Hard to make interpretations about the diel cycle of these data since they are so low. Which begs the question,

why are they so low at this site? Diel cycles of GOM at AI and TF are consistent with each other with an afternoon maximum, thus the statement on line 17 of Page 10 is misleading. Also, time axes in Figure 2 are not perfectly consistent for GOM and GEM. Please fix. Line 10, page 11, without some discussion of instrument intercomparison between the 3 sites, we cannot tell whether a GEM difference of 8% or 12% is simply due to the Tekran or is a real difference. Line 20, Page 11, the elevation of PM is 700 m asl, but this site is not a mountain peak and thus cannot be above the nocturnal boundary layer consistently. There may be more replenishment of GOM at this site, but again, the levels are super low and as such not much interpretation can be made of the GOM data at this site. In general, I feel the measurements from PM are of little value to the paper. The AI measurements are of greatest value since they are much higher and also are in the MBL where it appears that Br chemistry probably dominates. I would focus more on the model-measurement comparison at this site and less so on the comparison with the PM data.
* * *

---

## Referee Comment (RC2) · Anonymous Referee #2 · 19 Feb 2016

The authors present results of a box-model simulation of Hg chemistry at three sites in southern New Hampshire, USA. The sites are located in different environments (marine, coastal, and elevated), which allows the authors to examine the similarities and differences in Hg chemistry in these environments. The authors conclude that Br and BrO dominate Hg oxidation during the day and $H_2O_2$ at night at the marine site, while $O_3$ and OH are dominant at the coastal and inland sites. I found the comparison in Hg chemistry between the sites interesting. Atmospheric Hg chemistry remains one of the least understood processes controlling Hg cycling in the environment. Studies like this that use models to interpret in situ Hg observations in different environments are necessary to fully characterize the oxidation of Hg in the atmosphere. However, I

have a number of major concerns that the authors should address to make their study convincing.

Major comments: 1) The authors examine the oxidation of GEM with the set of gas-phase reactions listed in Table 1. There is high uncertainty in these reaction rates, up to a factor of 10 for reactions of GEM with Br and BrO. The recent review by Ariya et al. (2015) has a compilation of all previously reported estimates for GEM oxidation reaction rates. The authors perform one sensitivity study addressing the uncertainty in the GEM+O3 rate, but seem to ignore the uncertainty in the remaining reactions. A discussion of the effect of these uncertainties are necessary before any conclusion can be reached about the dominant GEM oxidation pathways at the studies sites.

2) How were the concentrations of the species that weren't measured set? How were the concentrations of Br/Cl/I species determined at the three sites? The authors briefly mention this in Section 3.4.1. This is a key aspect of the study and should be discussed in detail in Section 2.

3) The authors performed several sensitivity studies with the box-model by varying different physical and chemical parameters. However, these sensitivity studies seem out of place. The authors do not specify why they chose to vary the parameters listed in Table 3, and not others. Secondly, the presentation of the results of the sensitivity studies is not thorough. There is no discussion of how the results of the sensitivity studies affect the overall conclusions. Section 3.3.3 addresses Br chemistry in the MBL. I do not think this fits in this study, considering that there were no BrO measurements that could be used to compare with the model results.

4) Hg2+ reduction reactions were included in the model. Is reduction an important control on GOM mixing ratios? Some discussion of the effect of the reduction pathways on GOM and PBM would be valuable. The Tekran 2537/1130/1135 typically measures GEM, RGM, and PBM. The PBM measurements are not discussed in the manuscript. Can the PBM measurements be used to constrain the reduction rates?

5) In Section 1, the authors briefly discuss previous studies of Hg chemistry by Hedge-cock et al., Holmes et al., and Wang et al. The present study of Ye at al. is very similar to these previous studies. All of them examine the diurnal cycle of oxidized Hg in the mid-latitude marine boundary layer using a box-model. The authors should include a discussion of how their results compare with the findings in these previous studies.

Minor comments: Page 4, line 9: "Hg in the MBL cycles differently in coastal or inland areas." The difference needs to be expanded upon as this is directly related to the present study. How is the cycle different?

Page 5, line 23: Please add a list of reactions and their rates as a supplement, given that a few of the reactions do not seem to follow the JPL Report #17 recommendations.

Page 6, line 11: Please include a table with the reaction rates and references for the aqueous-phase reactions.

Page 7, line 1: Not all previous modeling studies have used simple approximations. The model of Hedgecock et al. (2003, 2005) uses detailed MBL chemistry.

Page 7, line 11: How are the wind speed measurements used in the box-model?

Page 7, line 19: "...were set to be constant during a simulation." Please specify the length of a simulation. Was it one day or one hour?

Page 9, line 20: It would be interesting to see how the source regions of the air masses at the three sites differed. The backtrajectories for only the AI site are discussed in the text.

Page 10, line 21: The LOD for the GOM observations was 0.1 ppqv at AI, but it appears to be much lower at PM. Figure 2 shows most GOM observations at PM below 0.05 ppqv. Please specify the LOD for GOM at PM.

Page 12, line 12: HgO is considered a GOM species here, although the authors state in the Introduction (page 3, line 22) that "a consensus has emerged that GEM+O3

reaction most likely occurs with solid-phase products..."

Page 12, line 19, I was surprised not to see HgBrNO2 as one of the more abundant GOM species. I expected HgBrNO2 to be produced faster than HgBrOOH and Hg-BrOBr, given the typically higher concentrations of NO2.

Page 13, line 14: Why were HgO and Hg(OH)2 more sensitive than halogenated GOM species to gas-particle partitioning?

Page 24, line 25: I do not see an order of magnitude difference between the peaks in Figure 5. I see a factor of 2-3 difference.

Page 19, lines 12 onwards: Could entrainment from the free troposphere explain this inverse relation between GOM and RH? Entrainment from the free troposphere was not treated explicitly, yet the boundary layer height at the TF site varied diurnally. Was it assumed that this entrainment does not change GOM mixing ratios in the boundary layer?

Page 21, line 29: "Clearly, the under-estimation case occurred under the strongest Bermuda High influence..." It isn't clear to me. Can the authors explain a bit more why it is the influence of the Bermuda High, and not just a transient high-pressure system?

Page 23, line 7: "It was hypothesized that..." This was not substantiated in the study, and does not belong in the conclusions.

Page 23, line 24: The authors allude to problems in GOM measurements using the Tekran instrument. If the measured GOM is indeed biased low by a factor of 2 or 3 under certain conditions, how does it affect this study's conclusions? This is important and needs to be discussed in a little more detail.

Page 23: The authors should also point out to the reader that, in the absence of speciated measurements of oxidized Hg compounds, the results of a modeling study cannot be used to conclusively identify the dominant oxidants of Hg in the atmosphere.

Table 2: Please include the standard deviation of the observed variables.

Figure 1: Please add some geographical context to the map. May be show the latitude/longitude girds, and the land/ocean boundary.

Figure 10: The backtrajectories suggest strong regional influence at the AI site. Can the authors reconcile this with their assumption for the box-model that regional transport is negligible?

Technical comments: Page 1, line 14: May be the title can specify that the study focuses on the summertime.

Page 1, line 14: The term Hg(II) is not needed here.

Page 3, line 7-8. "GOM and PBM are...subject to dry and wet deposition..." GEM is also subject to dry deposition.

Page 3, line 27: The sentence starting with "In the MBL..." needs to be rephrased for clarity.

Page 5, line 10: "...initial GEM mixing ratios...were set to be constant mimicking GEM emission flux." This sentence is unclear and should be reworded. I think removing the clause "and were set..." may help.

Page 10, line 4: FB is fractional bias.

Page 11, line 26: Reference to Section 3.2.2. Should this refer to Section 3.4?

Page 14, line 15: It seems TF_AIdry, PM_AIdry, TF_AIaero, PM_AIaero are not discussed any further. It would be better to not introduce them here.

Page 17, line 1: I think it would be more appropriate to place the model evaluation section before the sensitivity studies.

Page 19, line 27: "It was thus hypothesized that certain processes..." This sentence is vague. Please reword.

Page 23, line 15: "The updated chemical mechanism largely improved GOM simulations...". Improved with respect to what?

Figure 2: What do the "filled circles" represent? Expand the site abbreviations in the caption. The font size is too small.

Figure 3: Font size is too small. In the caption: do the bars "represent" the range of simulated GOM?

Figure 4: Please change "Other_RGM" to "Other GOM species". Please increase font size. It is hard to distinguish between the lines in Figure 3(a).

Figure 8: Caption: "("Observed", red, "Simulated", triangle)". Please correct typographical error.

Figure 9: It is difficult to distinguish between the Simulated_under-estimated, Simulated_matching, and Simulated_over-estimated lines. It would be also be helpful to maintain consistency between the figures in what is represented by the error bars.

---

## Referee Comment (RC3) · Anonymous Referee #3 · 25 Feb 2016

This paper uses a box model to study the controlling processes of GEM oxidation (or GOM formation) at different types of surface sites, and provides new and important information on the chemistry mechanisms of mercury that might occur in the real atmosphere. It fits well into the scope of ACP. I recommend the paper for publication after addressing the following comments. A major comment is that the box model simulation results should be compared against the measurements of PBM mixing ratios at these sites. This would help the interpretation of some controlling processes such as gas-particle partitioning in the model. Another general comment is that a more detailed description of the box model set up should be given in the paper, for example the exchange of GOM between the free troposphere and the boundary layer. A schematic

can be very helpful for the readers to understand which processes are discussed in the model. The third general comment is that the paper discusses the importance of different oxidized mercury forms through their oxidation pathways. I suggest the authors also discuss the stability of these oxidized forms in the real atmosphere in the particular environment of each site. Specific comments: 1. throughout the paper: the use of the word "case" in this paper may confuse its readers, as it refers to both different observational days and different model simulations. For example, in page 9, section 2.2 "Case selection", and in page 13, section 3.3 "Sensitivity analysis". 2. Title: it would be better if the full expression of GEM (i.e. gaseous elemental mercury) is given in the title. 3. Page 4, line 25. Can the authors describe which parameter is used to account for entrainment from the free troposphere? 4. Page 5, line 12. I do not understand why the "GEM mixing ratios ... are set to be constant mimicking GEM emission flux". What does this mean in the model? 5. Page 6, line 8. The numbers of reactions are incorrect. 6. Page 6, lines 11-18. The reaction constants for these aqueous Hg reactions should be given either in the main text or in the supplement. Also, I speculate these reactions are also highly uncertain. Do the authors consider the uncertainties associated with them? 7. Page 12, lines 1-9. These several sentences are confusing. At first, it is mentioned that "the patterns of diurnal variation were similar at the three sites". Then, it is said that "PM showed negligible diurnal variation". I suggest that a statistical method is used to quantitatively detect the diurnal patterns at all the sites. 8. Table 2: How uncertain are the simulated [Br] at TF and PM? What is the major source of [Br]? How is the concentration of Br2 set in the box model? In addition, are the boundary layer heights at AI and PM set to be constant? Do the authors expect any diurnal variations of the boundary layer height? 9. Figures: The figures throughout the paper should use a consistent way of uncertainty quantification, probably being consistent with the statistical method used for the observations (Figure 1). In the current paper, min-max, standard deviation, and box-whiskers all exist making the readers difficult to compare the uncertainties among these figures. In addition, I suggest the authors merge Figures 2, 3, and 8.

---

## Author Comment (AC1) · 16 May 2016

**Reviewer #1:**

This paper describes a model-measurement comparison study focusing on gaseous oxidized Hg (GOM) in the atmosphere at 3 sites in New Hampshire (marine, coastal, and inland). The main motivation for this study is to use updated reaction mechanisms and physical processes that control concentrations of GOM in the atmosphere and see if the model can reasonably reproduce the observations. There are many gaps in the understanding of what controls GOM in the atmosphere and these gaps are important to close since this species is readily wet and dry deposited and contributes to the burden of Hg accumulation in aquatic and terrestrial ecosystems. The photochemical Hg box model used in this work represents the state of the art with many updated reaction mechanism and physical processes. The sensitivity testing of the model output to changing chemical and physical parameters is very good. The model reproduces the observations reasonably well, the most notable agreement is that the differing overall GOM measured concentrations between sites (AI > TF > PM) is reproduced in the model. The model also gives some indication of which chemical species comprise GOM at the different sites. This is new and valuable information if it can be corroborated by measurements.

We thank the reviewer for their thoughtful, constructive comments and suggestions. The manuscript has been revised carefully. Below we addressed the review point by point.

The main issue overall I have with the paper is that there is little discussion of the GOM measurements made at the 3 sites. At the end of the paper the authors acknowledge the latest papers suggesting that GOM may be systematically underestimated by the Tekran methods, but in the paper the authors do not indicate if there are any potential measurement biases in the data from the 3 sites, and if so how these might change the conclusions reached from the model-measurement comparison. This is a fundamental weakness of such a study that uses measurements with a high degree of uncertainty to validate (or compare to) a model that is also uncertain. That limitation aside, however, there is a great deal of value in such a comparison, and I find this paper to be largely acceptable as-is.

As the reviewer pointed out, the potential uncertainty in ambient Hg measurements especially GOM is a consensus in the community at large. Recent laboratory experiments and reviews (Lyman et al., 2010; Jaffe et al., 2014; McClure et al., 2014; Huang and Gustin, 2015; Gustin et al., 2015) reported $O_3$ and relative humidity (RH) interferences on mercury halides for KCl-coated denuder, which was a part of Tekran 1130 unit used for GOM field measurements commonly in the community as well as the observations of this study. Huang and Gustin (2015) suggested a linear relationship between RH and GOM loss (in %) in GOM measurements, i.e., RH = 0.63 GOM loss % + 18.1, $r^2 = 0.49$, $p < 0.01$, over a RH range of 21% - 62%. In our GOM measurements, the interferences of RH at our three sites should have largely been eliminated since we used a custom-built refrigerator assembly and a canister of drierite to cool and dry air streams before entering into the 1130 pump module (Sigler et al., 2009). As a result, the RH of air streams was kept < 25%, therefore the upper limit of GOM loss caused by RH was < 10% using Huang and Gustin (2015)'s equation.

With regard to $O_3$ interference, the experimental study (Lyman et al., 2010) showed 3 to 37% reduction on the collection efficiency of $HgCl_2$, and the proposed reaction was $HgCl_2 + 2O_3 \rightarrow Hg^0 + 2O_2 + ClO$. However, a quantitative range of the bias caused by $O_3$ in field GOM measurements was yet derived (Lyman et al., 2010). Huang et al. (2013) showed lower collection

efficiency of KCl denuders compared to nylon membrane and the cation exchange membrane for $HgBr_2$, $HgCl_2$, HgO, $HgSO_4$, and $Hg(NO_3)_2$ in laboratory tests. However, for field measurements (Huang et al., 2013; Gustin et al., 2013), since GOM and PBM could not be distinguished from total reactive mercury using nylon membrane and cation exchange membrane chambers, the quantitative bias extent derived for total reactive mercury could not be directly used for GOM. Moreover, Huang et al. (2013) suggested that in their marine boundary layer site and highway impacted site, ambient GOM most likely existed in forms other than the laboratory tested species. Therefore, bias low GOM collection efficiency of KCl-coated denuders in field measurements remains speculative at this point.

Quality measurement data are used as ground truth for atmospheric Hg modeling studies, notwithstanding their limitation. Better instrumentation and/or solidly quantified bias for current instruments are in urgent need and are of essential importance to atmospheric Hg modeling. Nevertheless, even if models did not perfectly reproduced observations, the information derived from model simulations and sensitivity studies could provide insight into how the mechanisms work.

Abstract: I understand word count in limited, but there should be some indication of what is new or novel about the research. Which finding contributes to our understanding the best? It should be more than a list of observations, but rather some indication of why these observations matter.

The abstract was revised upon the reviewer's suggestion to reflect the findings of the study that are original.

Line 11, Page 3, provide reference(s) for this statement.

Reworded and references added.

Line 20 Page 6, "The ClO /BrO / IO radical cycles involve oxidation of Cl /Br / I radicals, photodissociation of $ClNO_2$ / $ClONO_2$ /$BrNO_2$ /$BrONO_2$, production from other halogen radicals, and sink reactions to reproduce Cl/Br/O radicals or other halides." What is meant by "reproduce"? Do you mean to reproduce the observations? What if those observations are very uncertain?

Changed to "calculate". No observations of Cl, Br and I radicals.

Line 15 Page 10, and Figure 2, there needs to be some discussion about the Tekran measurements. There were no mention of these in the methods section. Were the 3 instruments at the 3 locations operated in a consistent manner? Why such a large variation in GOM at AI, but very low levels at PM? The authors state the MDL for GOM is 0.1 ppqv yet most of the PM data in Figure 2 is < MDL. Hard to make interpretations about the diel cycle of these data since they are so low. Which begs the question, why are they so low at this site?

A brief discussion about the Tekran measurements was added upon the reviewer's suggestion (lines 193 - 200 in the revised manuscript).

GOM was collected over a 2-h sampling period at a rate of 10 L min$^{-1}$ using a speciation unit (Tekran 1130) installed upstream of the total gaseous mercury (TGM) analyzer. The instruments for the three sites were run and calibrated in the lab first and then operated at the sites in a consistent manner. The GOM detection limit for all three instruments were derived as ~0.1 ppqv, based on three times the standard deviation of the averaged blank (Sigler et al., 2009; Mao and Talbot, 2012). We added this information in section 2.1.2.

Pack Monadnock (PM) is a heavily forested, elevated, inland site, representing continental background conditions with nearly no marine influence. PM is not the only site with frequent below LOD measurements of GOM; in fact, similar levels of GOM have been reported from other background sites over the United States (Hall et al., 2006; Engle et al., 2010; Kolker et al., 2010; Choi et al., 2013).

Several possible reasons were proposed to explain significantly lower GOM mixing ratios at PM in comparison with the higher values at AI. First, the GEM oxidation at PM is not as active as that at AI due to a lack of halogen radicals. Second, the dry deposition velocity of GOM at PM (2 cm s$^{-1}$) was estimated a factor of 3 greater than that at AI (0.5 cm s$^{-1}$) using the values from Zhang et al. (2009, 2012). Third, the gas-particle partitioning process at PM was favorable for PBM formation, which could be conducive to a high loss rate of GOM. In fact, our model sensitivity runs suggested that the strong oxidation of GEM by O$_3$ at PM could lead to higher GOM mixing ratios (up to 4 ppqv) during daytime if the same gas-particle partitioning and dry deposition velocity that were used for AI were applied at PM (Figure 6 and Section 3.4.2 in revised manuscript). The simulated production and loss rates of GOM were on average 3.4 molecules cm$^{-3}$ s$^{-1}$ and 5.1 molecules cm$^{-3}$ s$^{-1}$, respectively, at 0.1 ppqv GOM. The production and loss were balanced out at 0.066 ppqv GOM. This suggests all the GOM produced from GEM oxidation at PM might have been lost rapidly via dry deposition and gas-particle partitioning. Moreover, PM would be in the residual layer at night, with air masses from the preceding daytime convective boundary layer where the GOM concentrations were typically below LOD.

Here we attached detailed information on GOM (also termed as RGM) measurements from Sigler et al. (2009):

"RGM is measured with a speciation unit (Tekran model 1130) consisting of a denuder and pump module installed upstream of the TGM analyzer. At TF and PM, the analyzer is housed in a temperature-controlled (~25°C) instrumentation shed. The denuder module is mounted on top of the shed at a height of approximately 5m. At AI, the denuder module is mounted at the top of a World War II-era observation tower (~20m), with the TGM analyzer installed inside the top floor.

The denuder module is attached to the pump module and TGM analyzer by a heated (50°C) umbilical line. The KCl-coated denuder strips out RGM during a predetermined sampling period while the TGM analyzer continuously measures Hg$^0$ (see Landis et al., 2002). Over the final 30 min of the sampling period, the denuder is flushed with zero air and heated to 500 °C so that the RGM is thermally absorbed and sampled (as Hg$^0$) by the TGM analyzer. Uncertainty of RGM measurements is high, especially at low levels, and we currently lack standard reference materials for calibration (Aspmo et al., 2005). To reduce uncertainty as much as possible, we strive for very low blanks. We measure RGM over a 2-h sampling period at a rate of 10 L min$^{-1}$, and with a detection limit of ~0.1 ppqv, based on three times the standard deviation of the averaged blank (e.g., 0.003±0.03 ppqv, n = 3626 at TF in 2007; Sigler et al., 2009).

Clean operation of the 1130 system is verified by flushing the system with zero air. Ideally the resultant mixing ratio during zero air flushes before and after denuder heating is 0 ppqv. To ensure clean operation, the denuders, denuder module glassware, impactor frits and sample filters are replaced and cleaned on a 10-day basis at TF and PM, and typically on a 2-3 week basis at AI. At TF and especially AI, high humidity may corrode zero air canisters, saturate soda lime and lead to poor blanks or enhance cartridge passivation. To minimize the potential of

moisture damage and improve blanks during desorption, the airstream leading into the 1130 pump module is cooled and dried using a custom-built refrigerator assembly and a canister of drierite. This system ensures that even when the drierite is exhausted, the relative humidity of the air entering the pump module is ~25% or less. At AI, humidity as well as sea salts led to high blanks during the first month of deployment in 2007. Addition of the refrigerator assembly along with replacement of an aging pump diaphragm on 9 August resulted in clean blank values (0 ppqv) on more than 80% of the RGM observations at AI for the remainder of the field campaign.

In our experience, mixing ratios of 0 ppqv are achieved for > 99% of zero air flushes after desorption and for >94% of zero air flushes immediately before desorption at both TF and PM. When a level of 0 ppqv is not achieved, a blank correction is made to the resultant mixing ratio based on the average value of measurements during zero air flushes before and after desorption."

Diel cycles of GOM at AI and TF are consistent with each other with an afternoon maximum, thus the statement on line 17 of Page 10 is misleading.

Deleted this sentence.

Also, time axes in Figure 2 are not perfectly consistent for GOM and GEM. Please fix.

Fixed.

Line 10, page 11, without some discussion of instrument intercomparison between the 3 sites, we cannot tell whether a GEM difference of 8% or 12% is simply due to the Tekran or is a real difference.

The instruments for the three sites were run and calibrated in the lab first and then operated at the sites in a consistent manner. Below information on the measurements at the three sites was added in the text (lines 193 - 197 in section 2.1.2 in the revised manuscript):

"For these three sites, the instruments were first run and calibrated in the laboratory and then operated at the sites in a consistent manner. GEM was measured at 5-min intervals and with a limit of detection (LOD) of ~5-10 ppqv (Mao et al., 2008), RGM was measured over a 2-h sampling period with a LOD of ~0.1 ppqv based on three times the standard deviation of the field blank values (Sigler et al., 2009; Mao and Talbot, 2012)."

Line 20, Page 11, the elevation of PM is 700 m asl, but this site is not a mountain peak and thus cannot be above the nocturnal boundary layer consistently. There may be more replenishment of GOM at this site, but again, the levels are super low and as such not much interpretation can be made of the GOM data at this site. In general, I feel the measurements from PM are of little value to the paper. The AI measurements are of greatest value since they are much higher and also are in the MBL where it appears that Br chemistry probably dominates. I would focus more on the model-measurement comparison at this site and less so on the comparison with the PM data.

We agree that the MBL data are most interesting to understanding Hg chemistry, whereas the GOM mixing ratios at PM appear to be too low for meaningful interpretation if we used the observational data at the site alone. However, in this study, in our opinion it is important to include PM measurements because it could provide a comparison of GOM mixing ratios from three very different environments. Pack Monadnock (PM) is a heavily forested, elevated, inland site, representing continental background conditions with nearly no marine influence. PM is not the only site with frequent below LOD measurements of GOM; in fact, similar levels of GOM have

been reported from other background sites over the United States (Hall et al., 2006; Engle et al., 2010; Kolker et al., 2010; Choi et al., 2012).

We agree that interpretation of GOM mixing ratios <LOD would not be of much value. The site comparison was limited largely for the sensitivity runs to determine the processes that could potentially result in such the observed site difference in GOM mixing ratios.

REFERENCE

Aspmo, K., Gauchard, P.-A., Steffen, A., Temme, C., Berg, T., Bahlmann, E., Banic, C., Dommergue, A., Ebinghaus, R., Ferrari, C., Pirrone, N., Sprovieri, F. and Wibetoe, G.: Measurements of atmospheric mercury species during an international study of mercury depletion events at Ny-Ålesund, Svalbard, spring 2003. How reproducible are our present methods?, Atmospheric Environment, 39(39 SPEC. ISS.), 7607–7619, 2005.

Choi, H.-D., Huang, J., Mondal, S. and Holsen, T. M.: Variation in concentrations of three mercury (Hg) forms at a rural and a suburban site in New York State, Sci. Total Environ., 448, 96–106, doi:10.1016/j.scitotenv.2012.08.052, 2013.

Engle, M. A., Tate, M. T., Krabbenhoft, D. P., Schauer, J. J., Kolker, A., Shanley, J. B. and Bothner, M. H.: Comparison of atmospheric mercury speciation and deposition at nine sites across central and eastern North America, J. Geophys. Res., 115(D18), D18306, doi:10.1029/2010JD014064, 2010.

Gustin, M. S., Amos, H. M., Huang, J., Miller, M. B. and Heidecorn, K.: Measuring and modeling mercury in the atmosphere: a critical review, Atmos. Chem. Phys., 15(10), 5697–5713, doi: 10.5194/acp-15-5697-2015, 2015.

Gustin, M. S., Huang, J., Miller, M. B., Peterson, C., Jaffe, D. A., Ambrose, J., Finley, B. D., Lyman, S. N., Call, K., Talbot, R., Feddersen, D., Mao, H. and Lindberg, S. E.: Do we understand what the mercury speciation instruments are actually measuring? Results of RAMIX, Environmental Science and Technology, 47(13), 7295–7306, 2013.

Hall, B. D., Olson, M. L., Rutter, A. P., Frontiera, R. R., Krabbenhoft, D. P., Gross, D. S., Yuen, M., Rudolph, T. M. and Schauer, J. J.: Atmospheric mercury speciation in Yellowstone National Park, Science of The Total Environment, 367(1), 354–366, doi:10.1016/j.scitotenv.2005.12.007, 2006.

Huang, J. and Gustin, M. S.: Uncertainties of Gaseous Oxidized Mercury Measurements Using KCl-Coated Denuders, Cation-Exchange Membranes, and Nylon Membranes: Humidity Influences, Environ. Sci. Technol., 49(10), 6102–6108, doi:10.1021/acs.est.5b00098, 2015.

Huang, J., Miller, M. B., Weiss-Penzias, P. and Gustin, M. S.: Comparison of gaseous oxidized Hg measured by KCl-coated denuders, and nylon and cation exchange membranes, Environmental Science and Technology, 47(13), 7307–7316, 2013.

Jaffe, D. A., Lyman, S., Amos, H. M., Gustin, M. S., Huang, J., Selin, N. E., Levin, L., ter Schure, A., Mason, R. P., Talbot, R., Rutter, A., Finley, B., Jaeglé, L., Shah, V., McClure, C., Ambrose, J., Gratz, L., Lindberg, S., Weiss-Penzias, P., Sheu, G.-R., Feddersen, D., Horvat, M., Dastoor, A., Hynes, A. J., Mao, H., Sonke, J. E., Slemr, F., Fisher, J. A., Ebinghaus, R., Zhang, Y. and Edwards,

G.: Progress on Understanding Atmospheric Mercury Hampered by Uncertain Measurements, Environ. Sci. Technol., 48(13), 7204–7206, doi:10.1021/es5026432, 2014.

Kolker, A., Olson, M. L., Krabbenhoft, D. P., Tate, M. T. and Engle, M. A.: Patterns of mercury dispersion from local and regional emission sources, rural Central Wisconsin, USA, Atmos. Chem. Phys., 10(10), 4467–4476, doi:10.5194/acp-10-4467-2010, 2010.

Landis, M. S., Stevens, R. K., Schaedlich, F. and Prestbo, E. M.: Development and characterization of an annular denuder methodology for the measurement of divalent inorganic reactive gaseous mercury in ambient air, Environ. Sci. Technol., 36(13), 3000–3009, 2002.

Lyman, S. N., Jaffe, D. A. and Gustin, M. S.: Release of mercury halides from KCl denuders in the presence of ozone, Atmos. Chem. Phys., 10(17), 8197–8204, doi:10.5194/acp-10-8197-2010, 2010.

Mao, H., Talbot, R. W., Sigler, J. M., Sive, B. C. and Hegarty, J. D.: Seasonal and diurnal variations of Hg° over New England, Atmospheric Chemistry and Physics, 8(5), 1401–1421, 2008.

Mao, H. and Talbot, R.: Speciated mercury at marine, coastal, and inland sites in New England-Part 1: Temporal variability, Atmospheric Chemistry and Physics, 12(11), 5099–5112, 2012.

McClure, C. D., Jaffe, D. A. and Edgerton, E. S.: Evaluation of the KCl Denuder Method for Gaseous Oxidized Mercury using HgBr2 at an In-Service AMNet Site, Environ. Sci. Technol., 48(19), 11437–11444, doi:10.1021/es502545k, 2014.

Sigler, J. M., Mao, H. and Talbot, R.: Gaseous elemental and reactive mercury in Southern New Hampshire, Atmos. Chem. Phys., 9(6), 1929–1942, doi:10.5194/acp-9-1929-2009, 2009.

Zhang, L., Wright, L. P. and Blanchard, P.: A review of current knowledge concerning dry deposition of atmospheric mercury, Atmospheric Environment, 43(37), 5853–5864, 2009.

Zhang, L., Blanchard, P., Johnson, D., Dastoor, A., Ryzhkov, A., Lin, C. J., Vijayaraghavan, K., Gay, D., Holsen, T. M., Huang, J., Graydon, J. A., Louis, V. L. S., Castro, M. S., Miller, E. K., Marsik, F., Lu, J., Poissant, L., Pilote, M. and Zhang, K. M.: Assessment of modeled mercury dry deposition over the Great Lakes region, Environmental Pollution, 161, 272–283, 2012.

---

## Author Comment (AC2) · 16 May 2016

**Reviewer #2:**

The authors present results of a box-model simulation of Hg chemistry at three sites in southern New Hampshire, USA. The sites are located in different environments (marine, coastal, and elevated), which allows the authors to examine the similarities and differences in Hg chemistry in these environments. The authors conclude that Br and BrO dominate Hg oxidation during the day and $H_2O_2$ at night at the marine site, while $O_3$ and OH are dominant at the coastal and inland sites. I found the comparison in Hg chemistry between the sites interesting. Atmospheric Hg chemistry remains one of the least understood processes controlling Hg cycling in the environment. Studies like this that use models to interpret in situ Hg observations in different environments are necessary to fully characterize the oxidation of Hg in the atmosphere. However, I have a number of major concerns that the authors should address to make their study convincing.

We thank the reviewer for their detailed, thoughtful, constructive comments and suggestions. The manuscript has been revised carefully. Below we addressed the review point by point.

Major comments: 1) The authors examine the oxidation of GEM with the set of gas phase reactions listed in Table 1. There is high uncertainty in these reaction rates, up to a factor of 10 for reactions of GEM with Br and BrO. The recent review by Ariya et al. (2015) has a compilation of all previously reported estimates for GEM oxidation reaction rates. The authors perform one sensitivity study addressing the uncertainty in the GEM+$O_3$ rate, but seem to ignore the uncertainty in the remaining reactions. A discussion of the effect of these uncertainties are necessary before any conclusion can be reached about the dominant GEM oxidation pathways at the studies sites.

The major oxidation reactions of GEM are GEM + $O_3$ and GEM + Br in our box model. For the GEM + Br reaction, Ariya et al. (2002) yielded a rate constant of $3.2 \times 10^{-12}$ cm$^3$ molecule$^{-1}$ s$^{-1}$ using a relative rate method. However, Ariya et al. (2002) used one single rate reference only, which largely limited the accuracy of their results (Hynes et al. 2009). Moreover, large amounts of cyclohexane (an OH scavenger) used in Ariya et al. (2002)'s experiment may lead to an enhancement in the absorption of reactants on the cell walls (Hynes et al. 2009). A number of studies (Spicer et al. 2002; Donohoue et al. 2006; Sumner et al. 2011; Subir et al., 2011; Goodsite et al., 2004, 2012) showed a narrow range of $(3.0 - 6.4) \times 10^{-13}$ cm$^3$ molecule$^{-1}$ s$^{-1}$ for the rate coefficient of GEM + Br, from which we used a temperature dependent rate coefficient of $3.7 \times 10^{-13}(T/298)^{-2.76}$ cm$^3$ molecule$^{-1}$ s$^{-1}$ from Goodsite et al. (2004; 2012). To further investigate the GEM + Br rate coefficient sensitivity on GOM simulation, we added a new sensitivity scenario using Ariya et al. (2002) rate coefficient (section 3.4.1 in the revised manuscript). As a result, using the greater rate coefficient of Ariya et al. (2002) produced a factor of 3 or higher GOM mixing ratios than the base scenario.

We added the following discussion on the effect of reaction kinetics uncertainties on model simulations in section 3.4.1 (lines 522 - 527) of revised manuscript:

"Using a slower rate coefficient of GEM + $O_3$ (Hall, 1995) had similar effects as not including the GEM + $O_3$ reaction, i.e. decreasing GOM mixing ratios, especially at nighttime, and brominated GOM species becoming dominant. The GEM + OH reaction was not as important as GEM + $O_3$ or Br. The use of a higher GEM + Br rate coefficient derived from the study by Ariya et al. (2002) caused more than a factor of 3 higher GOM and PBM resulting in overestimated GOM for most cases."

2) How were the concentrations of the species that weren't measured set? How were the concentrations of Br/Cl/I species determined at the three sites? The authors briefly mention this in Section 3.4.1. This is a key aspect of the study and should be discussed in detail in Section 2.

For species that were not measured, we use the chemical mechanism to calculate their concentrations. Initial concentrations of most unmeasured species were set as the values in similar environments from the literature if available. Br/Cl/I concentrations were all calculated from the model given initial concentrations of 1 pptv (e.g. Finley et al., 2008; except for AI) for $Br_2$, $Cl_2$, and $I_2$ species. At AI, we set the $Br_2$ concentration to be constant during simulations and used Saiz-Lopez et al. (2006)'s values to constrain [BrO]. At TF and PM, the initial concentrations of $Br_2$, $Cl_2$, and $I_2$ were not sensitive factors for the simulated concentrations of Br/Cl/I, because during the simulations, $Br_2$, $Cl_2$, and $I_2$ were rapidly depleted without sources in inland environments.

We have added such information in section 2.1.2 (lines 203 - 207 in the revised manuscript) as follows:

"Br/Cl/I concentrations were all calculated from the model given initial concentrations of 2 pptv (Finley et al., 2008; except for AI) for $Br_2$, $Cl_2$, and $I_2$ species. At AI, the $Br_2$ initial concentration was set to be constant during simulations and used Saiz-Lopez et al. (2006)'s values to constrain [BrO]. Detailed information can be found in Section 3.3.1."

3) The authors performed several sensitivity studies with the box-model by varying different physical and chemical parameters. However, these sensitivity studies seem out of place. The authors do not specify why they chose to vary the parameters listed in Table 3, and not others. Secondly, the presentation of the results of the sensitivity studies is not thorough. There is no discussion of how the results of the sensitivity studies affect the overall conclusions. Section 3.3.3 addresses Br chemistry in the MBL. I do not think this fits in this study, considering that there were no BrO measurements that could be used to compare with the model results.

The ranges of parameters in sensitivity studies were based upon the varying range of each parameter from observations and the literature. The liquid water content range was derived from Hedgecock et al. (2003). The temperature range was based on the magnitude of observed average temperature diurnal cycles. We added such information in section 3.4.1 of revised manuscript (lines 487 - 490).

More discussion on the effect of these sensitivity tests was added (lines 518 - 530 in the revised manuscript):

"In summary, the parameters used in gas-particle partitioning processes including solar radiation values, temperature, and the rate coefficients of major GEM oxidation reaction, could all affect simulated GOM mixing ratios but with varying degrees. Aerosol properties were suggested to play a very important role in the partitioning of ambient GOM and PBM species and thus should be better represented in future Hg model simulation studies. Using a slower rate coefficient of $GEM + O_3$ (Hall, 1995) had similar effects as not including the $GEM + O_3$ reaction, i.e. decreasing GOM mixing ratios, especially at nighttime, and brominated GOM species becoming dominant. The $GEM + OH$ reaction was not as important as $GEM + O_3$ or Br. The use of a higher $GEM + Br$ rate coefficient derived from the study by Ariya et al. (2002) caused more than a factor of 3 higher GOM and PBM concentrations resulting in overestimated GOM for most cases. GOM and PBM production appeared to favor lower temperature during daytime and higher temperature at night,

and simulated GOM concentrations were not as sensitive to temperature change as to solar radiation and gas-particle partitioning."

Regarding section 3.3.3 (section 3.4.3 in the revised manuscript), in our opinion, this is one of the original contributions this study offers. Considering the importance of halogen chemistry in Hg cycling, we think halogen chemistry needs to be interactive with Hg chemistry. Constraining the BrO simulations using the observations from Saiz-Lopez et al. (2006), our box model results suggested that Br and BrO are two key compounds in determining GOM mixing ratios in the MBL. Section 3.4.3 in the revised manuscript includes theoretical analysis and discussion of the important bromine reactions that could affect Br and BrO simulations, which has vital importance in this study and can provide guidance for future Hg studies.

4) $Hg^{2+}$ reduction reactions were included in the model. Is reduction an important control on GOM mixing ratios? Some discussion of the effect of the reduction pathways on GOM and PBM would be valuable. The Tekran 2537/1130/1135 typically measures GEM, RGM, and PBM. The PBM measurements are not discussed in the manuscript. Can the PBM measurements be used to constrain the reduction rates?

A table with aqueous Hg reactions used in our model was added as Table S1.

Aqueous Hg reduction is one of the major sources of GEM in the atmosphere. Therefore, aqueous Hg reactions is supposedly a factor controlling GEM mixing ratios and further influence GOM mixing ratios. However, in this study aqueous Hg reduction was not an important control on GOM simulations. This is because GEM mixing ratios in the model were fixed using observed values. The uncertainties associated with aqueous Hg reactions would not influence GEM mixing ratios and therefore have minor effects on simulated GOM mixing ratios.

It is true that high quality GOM and PBM measurements would be of great help for modelling studies to evaluate the schemes such as gas-particle partitioning process as well as to constrain the aqueous reduction rate. However, the inlet of the Tekran speciation sampling system had an elutriator inlet with an acceleration jet to remove aerosols > 2.5 μm so that only PBM on fine particles was measured. The PBM calculated from the box model does not include size fractionation, thus Tekran $PBM_{2.5}$ measurements could not be used to constrain our simulations and the reduction rate.

5) In Section 1, the authors briefly discuss previous studies of Hg chemistry by Hedgecock et al., Holmes et al., and Wang et al. The present study of Ye at al. is very similar to these previous studies. All of them examine the diurnal cycle of oxidized Hg in the mid-latitude marine boundary layer using a box-model. The authors should include a discussion of how their results compare with the findings in these previous studies.

Section 1 was revised and expanded to reflect the aspects of this study that differentiate it from previous studies.

Minor comments: Page 4, line 9: "Hg in the MBL cycles differently in coastal or inland areas." The difference needs to be expanded upon as this is directly related to the present study. How is the cycle different?

The major differences of Hg cycles between MBL and coastal or inland areas are reflected in the magnitude and speciation of GOM, which are due to different chemical, meteorological and atmospheric conditions such as halogen radical mixing ratios, boundary layer height, and

atmospheric particles size and properties. More detailed discussion can be found in section 3.1 and section 3.4.2 in the revised manuscript.

Page 5, line 23: Please add a list of reactions and their rates as a supplement, given that a few of the reactions do not seem to follow the JPL Report #17 recommendations.

We have 424 reactions in total, which is too many to be included in the publication. We would be happy to provide the reactions upon request.

Halogen reactions listed in Table 4 were following the halogen chemistry reviews by Atkinson et al. (2004; 2008). We added this information in section 2.

Page 6, line 11: Please include a table with the reaction rates and references for the aqueous-phase reactions.

A table showing aqueous Hg reactions in our model was added as Table S1.

Page 7, line 1: Not all previous modeling studies have used simple approximations. The model of Hedgecock et al. (2003, 2005) uses detailed MBL chemistry.

The sentence has been revised.

Page 7, line 11: How are the wind speed measurements used in the box-model?

Wind speed measurements were used for case selection, not input for the box model. The text was revised to reflect this.

Page 7, line 19: "...were set to be constant during a simulation." Please specify the length of a simulation. Was it one day or one hour?

The length of a simulation is one hour. The sentence was revised to include this.

Page 9, line 20: It would be interesting to see how the source regions of the air masses at the three sites differed. The back trajectories for only the AI site are discussed in the text.

The back trajectories for the PM and TF sites (Fig. 1) showed air masses source regions. Air masses reaching PM originated from inland areas west to north of the site, while air masses at TF half came from northwestern to northern inland areas and half from the marine boundary layer. However, we did not find correlation between source regions and GOM mixing ratios at TF and PM. This is why the origin of air masses at the two sites was not discussed.

[Figure]

Figure 1. Clustered 24-hour backward trajectories of air masses in all cases at PM and TF.

Page 10, line 21: The LOD for the GOM observations was 0.1 ppqv at AI, but it appears to be much lower at PM. Figure 2 shows most GOM observations at PM below 0.05 ppqv. Please specify the LOD for GOM at PM.

The GOM detection limit for all three instruments were derived as ~0.1 ppqv, based on three times the standard deviation of the averaged blank (Sigler et al., 2009; Mao and Talbot, 2012). We added this information in section 2.1.2.

Page 12, line 12: HgO is considered a GOM species here, although the authors state in the Introduction (page 3, line 22) that "a consensus has emerged that GEM+$O_3$ reaction most likely occurs with solid-phase products..."

The experimental study by Pal and Ariya (2004) measured 1% of HgO produced by GEM + $O_3$ on an aerosol filter. Snider et al. (2008) showed HgO(s) production in their kinetic and product study. A theoretic study of Schroeder et al. (1998) suggested HgO would not exist as an isolated molecule at a decomposition temperature of +500 °C. However, the GEM + $O_3$ reaction and decomposition temperature (Schroeder et al., 1998) could also be impacted by the presence of other ambient gases (Snider et al., 2008; Gustin et al., 2013; Seigneur et al., 1994). Moreover, a recent study by Huang et al. (2013) observed gas-phase HgO using nylon and cation exchange membranes. Overall our knowledge about this reaction remains nebulous. We added this discussion in the introduction (section 1).

Page 12, line 19, I was surprised not to see $HgBrNO_2$ as one of the more abundant GOM species. I expected $HgBrNO_2$ to be produced faster than HgBrOOH and Hg-BrOBr, given the typically higher concentrations of $NO_2$.

In checking reactions forming $HgBrNO_2$, we found a mistake in $NO_x$ input. We should have fixed $NO_x$ concentrations in the input for the simulations but it was mistakenly left unfixed. In this revised version, we have rectified the mistake. As a result, the dominant brominated GOM species was changed to $HgBrNO_2$, and following with HgBrO; other brominated GOM species were negligible. However, Hg+Br reaction is so slow compared to further HgBr oxidation reactions that Hg+Br is the rate-limiting step for these two steps of reactions. Therefore, the change in the total GOM production was minor, and major conclusions remain unchanged (See Section 3.2 in the revised manuscript).

Page 13, line 14: Why were HgO and Hg(OH)$_2$ more sensitive than halogenated GOM species to gas-particle partitioning?

The difference between sensitivity of HgO/Hg(OH)$_2$ and halogenated GOM species to gas-particle partitioning was caused by higher molar mass of halogenated GOM species than HgO/Hg(OH)$_2$. When taken into calculations, compounds with smaller molar mass had a higher gas-to-particle rate based on the scheme described in section 2.1.3. The Henry's constant values of Hg(OH)$_2$ and halogenated GOM species are large enough to be not as sensitive as the molar mass of the compounds is to gas-to-particle partitioning.

Page 24, line 25: I do not see an order of magnitude difference between the peaks in Figure 5. I see a factor of 2-3 difference.

Corrected.

Page 19, lines 12 onwards: Could entrainment from the free troposphere explain this inverse relation between GOM and RH? Entrainment from the free troposphere was not treated explicitly, yet the boundary layer height at the TF site varied diurnally. Was it assumed that this entrainment does not change GOM mixing ratios in the boundary layer?

In this study, we selected clear-sky and calm wind conditions, usually accompanied by strong stability with a strong inversion layer at the top of the daytime convective PBL layer based on measurements from the literature (e.g., Hogan et al., 2009). Minimal entrainment at the top of the boundary layer was thus expected.

We agree with the reviewer's point that at TF, the GOM in the remnant layer could be mixed down to the surface in the morning when the boundary layer rises. The observed daytime GOM mixing ratio peak is around 0.8 ppqv, and the contribution of downward mixing from the remnant layer at TF was estimated by Mao et al. (2006) to be about ~23% in the time window of after sunrise and 10 am local time. Under such circumstances, the contribution from the preceding convective boundary layer to the morning GOM mixing ratios would at most be ~0.2 ppqv. Moreover, even though GOM in the remnant layer at night did not deposit to the surface, it could be lost by deposition to aerosols and via other unknown mechanisms. Taking these into consideration, that 0.2 ppqv contribution from the remnant layer would be the upper limit. As the day progresses and solar radiation gets stronger, the GOM mixing ratio is mostly driven by photochemical production.

Page 21, line 29: "Clearly, the under-estimation case occurred under the strongest Bermuda High influence..." It isn't clear to me. Can the authors explain a bit more why it is the influence of the Bermuda High, and not just a transient high-pressure system?

The under-estimation cases were 06/13/2008 and 08/22/2007, the meteorological conditions of these days were illustrated using the NCEP 1° x 1° meteorological reanalysis data (Fig. 2). The observed GOM concentrations peaked at 14:00 LT on 06/13/2008 and 16:00 LT on 08/22/2007 respectively. On 13 June 2008, the Bermuda high pressure system covered almost the entire eastern US coastline, where our sites are located. This high pressure system lasted 4 days (10 – 14 June 2008). On 22 August 2007, the continental part of the Bermuda high pressure system was over the southeastern US extending to the northeast. These lasting high pressure systems caused regional buildup pollutants, explaining the observed high mixing ratios of GOM in the two cases. We will include these figures in the supplemental material.

[Figure]

Figure 2. Geopotential height for a) 06/13/2008 08:00 EDT, b) 06/13/2008 14:00 EDT, c) 08/22/2007 14:00 EDT, and d) 08/22/2007 20:00 EDT at 850 hPa, the green star shows the location of TF site.

Page 23, line 7: "It was hypothesized that..." This was not substantiated in the study, and does not belong in the conclusions.

These hypotheses were developed based on the modeling and analysis work in the paper. The text was revised to reflect the logical steps to take to arrive at the hypotheses.

Page 23, line 24: The authors allude to problems in GOM measurements using the Tekran instrument. If the measured GOM is indeed biased low by a factor of 2 or 3 under certain conditions, how does it affect this study's conclusions? This is important and needs to be discussed in a little more detail.

Page 23: The authors should also point out to the reader that, in the absence of speciated measurements of oxidized Hg compounds, the results of a modeling study cannot be used to conclusively identify the dominant oxidants of Hg in the atmosphere.

The reviewer raised excellent points here. We agree that without measurements of speciated GOM, modeling results cannot be used to conclusively identify the dominant oxidants of Hg, as well as dominant GOM species in that matter, in the atmosphere. Indeed the potential uncertainty in ambient Hg measurements especially GOM is a major concern in the community. We had some discussion on the effect of uncertainty in GOM measurements on our interpretation of measurements data. With the reviewer's suggestion in mind, the discussion was expanded to discuss the potential effect of biased low GOM measurements on our conclusions in the last section (lines 623 - 640 in the revised manuscript).

That being said, it is unlikely to put any range on the bias of our GOM concentrations considering our own GOM measurements and the literature. Recent laboratory experiments and reviews (Lyman et al., 2010; Jaffe et al., 2014; McClure et al., 2014; Huang and Gustin, 2015; Gustin et al., 2015) reported $O_3$ and relative humidity (RH) interferences on mercury halides for KCl-coated denuder, which was a part of Tekran 1130 unit used for GOM field measurements commonly in the community as well as the observations of this study. Huang and Gustin (2015) suggested a linear relationship between RH and GOM loss (in %) in GOM measurements, i.e., RH = 0.63 GOM loss % + 18.1, $r^2$ = 0.49, p < 0.01, at RH range of 21 to 62%. In our GOM measurements, the interferences of RH at our sites should have largely been eliminated since we used a custom-built refrigerator assembly and a canister of drierite to cool and dry air streams before entering into the 1130 pump module (Sigler et al., 2009). As a result, the RH of air streams was kept < 25%, therefore the upper limit of GOM loss cause by RH was < 10% using Huang and Gustin (2015)'s equation.

With regard to $O_3$ interference, the experimental study (Lyman et al., 2010) showed 3 to 37% reduction on the collection efficiency of $HgCl_2$, and the proposed reaction was $HgCl_2$ + $2O_3$ → $Hg^0$ + $2O_2$ + ClO. However, the quantitative extent of the bias caused by $O_3$ in field GOM measurements was yet derived (Lyman et al., 2010). Huang et al. (2013) showed lower collection efficiency of KCl denuders compared to nylon membrane and the cation exchange membrane for $HgBr_2$, $HgCl_2$, HgO, $HgSO_4$, and $Hg(NO_3)_2$ in laboratory tests. However, for field measurements (Huang et al., 2013; Gustin et al., 2013), since GOM and PBM could not be distinguished from total reactive mercury using nylon membrane and cation exchange membrane chambers, the quantitative bias extent derived for total reactive mercury could not be directly used for GOM. Moreover, Huang et al. (2013) suggested that in their marine boundary layer site and highway impacted site, ambient GOM most likely existed in forms other than the laboratory tested species. Therefore, bias low GOM collection efficiency of KCl-coated denuders in field measurements remains speculative at this point.

Quality measurement data are used as ground truth for atmospheric Hg modeling studies, notwithstanding their limitation. Better instrumentation and/or solidly quantified bias for current instruments are in urgent need and are of essential importance to atmospheric Hg modeling. Nevertheless, even if models did not perfectly reproduced observations, the information derived from model simulations and sensitivity studies could provide insight into how the mechanisms work.

The discussion added in the Summary section is as follows (lines 623 - 640 in the revised manuscript):

"It should be noted that without measurements of speciated GOM, modeling results cannot be used to conclusively identify the dominant oxidants of Hg, as well as dominant GOM species in that matter, in the atmosphere. Indeed, the potential uncertainty in ambient Hg measurements especially GOM is a major concern in the community. That being said, it is unlikely to have a quantitative understanding of the bias of our GOM concentrations. Recent laboratory experiments and reviews (Lyman et al., 2010; Jaffe et al., 2014; McClure et al., 2014; Huang and Gustin, 2015; Gustin et al., 2015) reported $O_3$ and relative humidity (RH) interferences on mercury halides for KCl-coated denuder, the part of Tekran 1130 unit commonly used for GOM field measurements. As stated in Section 2, in our GOM measurement the RH effect was minimized by adding refrigeration to remove excess of water in the airsteam. $O_3$ interference and bias low GOM

collection efficiency of KCl-coated denuders were limited to a handful of GOM species in laboratory experiments and remain untested in field measurements. If the measured GOM concentrations were indeed biased low by a factor of 2 or 3 under certain conditions as previous studies speculated, the matching cases at AI and TF would be reduced from 50% of the total cases to 30%, and the model would potentially underestimate GOM concentrations in the remaining cases (70%) by a factor of 3 to 4. It is however hard to speculate the effect at PM since most GOM observations there were below the LOD. This suggested even greater unknowns in our understanding of Hg chemistry."

Table 2: Please include the standard deviation of the observed variables.

Added the standard deviation values for observed variables in Table 2.

Figure 1: Please add some geographical context to the map. May be show the latitude/ longitude girds, and the land/ocean boundary.

Plotted a new map for Figure 1 (Fig. 3 showing below) with latitude/longitude grids and land/ocean boundary showed.

[Figure]

Figure 3. New Hampshire site map: Appledore Island (marine), Thompson Farm (coastal), and Pack Monadnock (inland elevated).

Figure 10: The back trajectories suggest strong regional influence at the AI site. Can the authors reconcile this with their assumption for the box-model that regional transport is negligible?

The trajectories were used to identify the origin of the air mass reaching AI. GEM was long-lived enough to originate from the same source region of the air mass. However, GOM in the air masses did not necessarily originate from the same source region due to its short lifetime. Under the conditions of strong atmospheric stability as selected in this study, GOM would likely be in-situ, photochemically produced.

Technical comments: Page 1, line 14: May be the title can specify that the study focuses on the summertime.

Upon the reviewer's suggestion the title was changed to "Investigation of processes controlling summertime gaseous elemental mercury oxidation at mid-latitudinal marine, coastal, and inland sites".

Page 1, line 14: The term Hg(II) is not needed here.

Deleted.

Page 3, line 7-8. "GOM and PBM are...subject to dry and wet deposition..." GEM is also subject to dry deposition.

Added.

Page 3, line 27: The sentence starting with "In the MBL..." needs to be rephrased for clarity.

Revised.

Page 5, line 10: "...initial GEM mixing ratios...were set to be constant mimicking GEM emission flux." This sentence is unclear and should be reworded. I think removing the clause "and were set..." may help.

Revised.

Page 10, line 4: FB is fractional bias.

Changed.

Page 11, line 26: Reference to Section 3.2.2. Should this refer to Section 3.4?

In this sentence, we meant that the reasons of large variations of GOM daytime peaks between AI, TF, and PM. We have discussed this in Section 3.4.2 of revised manuscript. We have corrected this.

Page 14, line 15: It seems TF_AIdry, PM_AIdry, TF_AIaero, PM_AIaero are not discussed any further. It would be better to not introduce them here.

Deleted.

Page 17, line 1: I think it would be more appropriate to place the model evaluation section before the sensitivity studies.

Agreed and done.

Page 19, line 27: "It was thus hypothesized that certain processes..." This sentence is vague. Please reword.

Revised.

Page 23, line 15: "The updated chemical mechanism largely improved GOM simulations...". Improved with respect to what?

Revised to "The updated chemical mechanism largely improved the simulation of the magnitude and pattern of GOM diurnal variation at the coastal and inland sites."

Figure 2: What do the "filled circles" represent? Expand the site abbreviations in the caption. The font size is too small.

The figure was revised.

Figure 3: Font size is too small. In the caption: do the bars "represent" the range of simulated GOM?

The figure was revised and the font size was increased for better presentation. Now the bars represent standard deviations of simulated GOM.

Figure 4: Please change "Other_RGM" to "Other GOM species". Please increase font size. It is hard to distinguish between the lines in Figure 3(a).

Changed.

Figure 8: Caption: "("Observed", red, "Simulated", triangle)". Please correct typographical error.

Corrected.

Figure 9: It is difficult to distinguish between the Simulated_under-estimated, Simulated_matching, and Simulated_over-estimated lines. It would be also be helpful to maintain consistency between the figures in what is represented by the error bars.

We revised the figures and used error bars for standard deviation only.

REFERENCE

Ariya, P. A., Khalizov, A. and Gidas, A.: Reactions of gaseous mercury with atomic and molecular halogens: Kinetics, product studies, and atmospheric implications, Journal of Physical Chemistry A, 106(32), 7310–7320, 2002.

Atkinson, R., Baulch, D. L., Cox, R. A., Crowley, J. N., Hampson, R. F., Hynes, R. G., Jenkin, M. E., Rossi, M. J., Troe, J. and Wallington, T. J.: Evaluated kinetic and photochemical data for atmospheric chemistry: Volume IV – gas phase reactions of organic halogen species, Atmos. Chem. Phys., 8(15), 4141–4496, doi:10.5194/acp-8-4141-2008, 2008.

Donohoue, D. L., Bauer, D., Cossairt, B. and Hynes, A. J.: Temperature and pressure dependent rate coefficients for the reaction of Hg with Br and the reaction of Br with Br: A pulsed laser photolysis-pulsed laser induced fluorescence study, Journal of Physical Chemistry A, 110(21), 6623–6632, 2006.

Finley, B. D. and Saltzman, E. S.: Observations of $Cl_2$, $Br_2$, and $I_2$ in coastal marine air, J. Geophys. Res., 113(D21), D21301, doi:10.1029/2008JD010269, 2008.

Goodsite, M. E., Plane, J. M. C. and Skov, H.: A Theoretical Study of the Oxidation of $Hg^0$ to $HgBr_2$ in the Troposphere, Environmental Science and Technology, 38(6), 1772–1776, 2004.

Goodsite, M. E., Plane, J. M. C. and Skov, H.: Correction to A Theoretical Study of the Oxidation of $Hg^0$ to $HgBr_2$ in the Troposphere, Environ. Sci. Technol., 46(9), 5262–5262, doi:10.1021/es301201c, 2012.

Gustin, M. S., Amos, H. M., Huang, J., Miller, M. B. and Heidecorn, K.: Measuring and modeling mercury in the atmosphere: a critical review, Atmos. Chem. Phys., 15(10), 5697–5713, doi: 10.5194/acp-15-5697-2015, 2015.

Gustin, M. S., Huang, J., Miller, M. B., Peterson, C., Jaffe, D. A., Ambrose, J., Finley, B. D., Lyman, S. N., Call, K., Talbot, R., Feddersen, D., Mao, H. and Lindberg, S. E.: Do we understand what the mercury speciation instruments are actually measuring? Results of RAMIX, Environmental Science and Technology, 47(13), 7295–7306, 2013.

Hall, B.: The gas phase oxidation of elemental mercury by ozone, Water Air Soil Pollut., 80(1-4), 301–315, 1995.

Hedgecock, I. M., Pirrone, N., Sprovieri, F. and Pesenti, E.: Reactive gaseous mercury in the marine boundary layer: Modelling and experimental evidence of its formation in the Mediterranean region, Atmos.Environ., 37(SUPPL. 1), S41–S49, 2003.

Hogan, R. J., Grant, A. L. M., Illingworth, A. J., Pearson, G. N. and O'Connor, E. J.: Vertical velocity variance and skewness in clear and cloud-topped boundary layers as revealed by Doppler lidar, Q.J.R. Meteorol. Soc., 135(640), 635–643, doi:10.1002/qj.413, 2009.

Holmes, C. D., Jacob, D. J., Mason, R. P. and Jaffe, D. A.: Sources and deposition of reactive gaseous mercury in the marine atmosphere, Atmospheric Environment, 43(14), 2278–2285, 2009.

Huang, J. and Gustin, M. S.: Uncertainties of Gaseous Oxidized Mercury Measurements Using KCl-Coated Denuders, Cation-Exchange Membranes, and Nylon Membranes: Humidity Influences, Environ. Sci. Technol., 49(10), 6102–6108, doi:10.1021/acs.est.5b00098, 2015.

Huang, J., Miller, M. B., Weiss-Penzias, P. and Gustin, M. S.: Comparison of gaseous oxidized Hg measured by KCl-coated denuders, and nylon and cation exchange membranes, Environmental Science and Technology, 47(13), 7307–7316, 2013.

Hynes, A. J., Donohoue, D. L., Goodsite, M. E. and Hedgecock, I. M.: Our current understanding of major chemical and physical processes affecting mercury dynamics in the atmosphere and at the air-water/terrestrial interfaces, Mercury Fate and Transport in the Global Atmosphere: Emissions, Measurements and Models, (Journal Article), 427–457, 2009.

Jaffe, D. A., Lyman, S., Amos, H. M., Gustin, M. S., Huang, J., Selin, N. E., Levin, L., ter Schure, A., Mason, R. P., Talbot, R., Rutter, A., Finley, B., Jaeglé, L., Shah, V., McClure, C., Ambrose, J., Gratz, L., Lindberg, S., Weiss-Penzias, P., Sheu, G.-R., Feddersen, D., Horvat, M., Dastoor, A., Hynes, A. J., Mao, H., Sonke, J. E., Slemr, F., Fisher, J. A., Ebinghaus, R., Zhang, Y. and Edwards, G.: Progress on Understanding Atmospheric Mercury Hampered by Uncertain Measurements, Environ. Sci. Technol., 48(13), 7204–7206, doi:10.1021/es5026432, 2014.

Lyman, S. N., Jaffe, D. A. and Gustin, M. S.: Release of mercury halides from KCl denuders in the presence of ozone, Atmos. Chem. Phys., 10(17), 8197–8204, doi:10.5194/acp-10-8197-2010, 2010.

Mao, H. and Talbot, R.: Speciated mercury at marine, coastal, and inland sites in New England-Part 1: Temporal variability, Atmospheric Chemistry and Physics, 12(11), 5099–5112, 2012.

Mao, H., Talbot, R., Nielsen, C. and Sive, B.: Controls on methanol and acetone in marine and continental atmospheres, Geophys. Res. Lett., 33(2), L02803, doi:10.1029/2005GL024810, 2006.

McClure, C. D., Jaffe, D. A. and Edgerton, E. S.: Evaluation of the KCl Denuder Method for Gaseous Oxidized Mercury using $HgBr_2$ at an In-Service AMNet Site, Environ. Sci. Technol., 48(19), 11437–11444, doi:10.1021/es502545k, 2014.

Pal, B. and Ariya, P. A.: Studies of ozone initiated reactions of gaseous mercury: Kinetics, product studies, and atmospheric implications, Physical Chemistry Chemical Physics, 6(3), 572–579, 2004.

Saiz-Lopez, A., Shillito, J. A., Coe, H., and Plane, J. M. C.: Measurements and modelling of $I_2$, IO, OIO, BrO and $NO_3$ in the mid-latitude marine boundary layer, Atmos. Chem. Phys., 6, 1513–1528, doi:10.5194/acp-6-1513-2006, 2006.

Schroeder, W. H. and Munthe, J.: Atmospheric mercury - An overview, Atmospheric Environment, 32(5), 809–822, 1998.

Seigneur, C., Wrobel, J. and Constantinou, E.: A chemical kinetic mechanism for atmospheric inorganic mercury, Environmental Science and Technology, 28(9), 1589–1597, 1994.

Sigler, J. M., Mao, H. and Talbot, R.: Gaseous elemental and reactive mercury in Southern New Hampshire, Atmos. Chem. Phys., 9(6), 1929–1942, doi:10.5194/acp-9-1929-2009, 2009.

Snider, G., Raofie, F. and Ariya, P. A.: Effects of relative humidity and CO(g) on the $O_3$-initiated oxidation reaction of $Hg^0$(g): Kinetic & product studies, Physical Chemistry Chemical Physics, 10(36), 5616–5623, 2008.

Spicer, C. W., Satola, J., Abbgy, A. A., Plastridge, R. A. and Cowen, K. A.: Kinetics of Gas-Phase Elemental Mercury Reaction with Halogen Species, Ozone, and Nitrate Radical under Atmospheric Conditions, (Journal Article), 2002.

Subir, M., Ariya, P. A. and Dastoor, A. P.: A review of uncertainties in atmospheric modeling of mercury chemistry I. Uncertainties in existing kinetic parameters - Fundamental limitations and the importance of heterogeneous chemistry, Atmos.Environ., 45(32), 5664–5676, 2011.

Sumner, A. L., Spicer, C. W., Landis, M. S. and Stevens, R. K.: Kinetics of gaseous elemental mercury oxidation reactions under conditions of relevance to the atmosphere, Atmos.Environ., 2011.

Wang, F., Saiz-Lopez, A., Mahajan, A. S., Martín, J. C. G., Armstrong, D., Lemes, M., Hay, T. and Prados-Roman, C.: Enhanced production of oxidised mercury over the tropical Pacific Ocean: A key missing oxidation pathway, Atmospheric Chemistry and Physics, 14(3), 1323–1335, 2014.

---

## Author Comment (AC3) · 16 May 2016

**Reviewer #3:**

This paper uses a box model to study the controlling processes of GEM oxidation (or GOM formation) at different types of surface sites, and provides new and important information on the chemistry mechanisms of mercury that might occur in the real atmosphere. It fits well into the scope of ACP. I recommend the paper for publication after addressing the following comments.

We thank the reviewer for their thoughtful, constructive comments and suggestions. The manuscript has been revised carefully. Below we addressed the reviews point by point.

A major comment is that the box model simulation results should be compared against the measurements of PBM mixing ratios at these sites. This would help the interpretation of some controlling processes such as gas-particle partitioning in the model.

It is true that high quality GOM and PBM measurements would be of great help for modelling studies to evaluate the schemes such as gas-particle partitioning process as well as to constrain the aqueous reduction rate. However, the inlet of the Tekran speciation sampling system had an elutriator inlet with an acceleration jet to remove aerosols > 2.5 μm so that only fine PBM was measured. The PBM calculated from the box model does not include size fractionation, thus Tekran $PBM_{2.5}$ measurements could not be used to constrain our simulations and further constrain the reduction rate.

Another general comment is that a more detailed description of the box model set up should be given in the paper, for example the exchange of GOM between the free troposphere and the boundary layer. A schematic can be very helpful for the readers to understand which processes are discussed in the model.

We did not include a scheme to account for GOM exchange between the free troposphere and the boundary layer. Such exchange processes are highly parameterized, and location and time dependent. Including such processes could induce another major uncertainty in the model. In this study, we selected clear-sky and calm wind conditions, usually accompanied by strong stability with a strong inversion layer at the top of the daytime convective PBL layer based on measurements from the literature (e.g.Hogan et al., 2009). Minimal entrainment at the top of the boundary layer was thus expected.

The third general comment is that the paper discusses the importance of different oxidized mercury forms through their oxidation pathways. I suggest the authors also discuss the stability of these oxidized forms in the real atmosphere in the particular environment of each site.

This is a valid point. However, since properties of the oxidized forms remain largely unknown, we added a general discussion on the possible impact of different environments on speciation. The discussion added in the text is as follows in section 1 of revised manuscript:

"GOM concentrations and speciation could be impacted by meteorological conditions and chemical conditions in different environments. High solubility of GOM species, possible phase partitioning of HgO as discussed above could all be the reasons causing varying GOM speciation at different locations. For instance, the aerosol type, size distribution, and chemical composition varied largely between the MBL site and inland sites, which may lead to different gas-particle partitioning rates of GOM species."

Specific comments: 1. throughout the paper: the use of the word "case" in this paper may confuse its readers, as it refers to both different observational days and different model simulations. For example, in page 9, section 2.2 "Case selection", and in page 13, section 3.3 "Sensitivity analysis".

We have changed the word "case" in sensitivity studies to "scenario".

2. Title: it would be better if the full expression of GEM (i.e. gaseous elemental mercury) is given in the title.

Upon the reviewer's suggestion the title was changed to "Investigation of processes controlling summertime gaseous elemental mercury oxidation at mid-latitudinal marine, coastal, and inland sites".

3. Page 4, line 25. Can the authors describe which parameter is used to account for entrainment from the free troposphere?

As we responded above to a comment similar to this, we did not include a scheme to account for GOM exchange between the free troposphere and the boundary layer. Such exchange processes are highly parameterized, and location and time dependent. Including such processes could induce another major uncertainty in the model. In this study, we selected clear-sky and calm wind conditions, usually accompanied by strong stability with a strong inversion layer at the top of the daytime convective PBL layer based on measurements from the literature (e.g., Hogan et al., 2009). Minimal entrainment at the top of the boundary layer was thus expected.

4. Page 5, line 12. I do not understand why the "GEM mixing ratios ... are set to be constant mimicking GEM emission flux". What does this mean in the model?

Revised to "the initial GEM mixing ratios along with a list of compounds (Table 2) in the model were obtained from observations in three different environments are were set to be constant during simulations". The theory behind the fixed input concentrations of GEM among a number of other compounds is that a box model simulates the concentrations of short-lived compounds reaching an instantaneous chemical steady state, and for the time scales of such instants, the chemicals such as GEM are long-lived enough to maintain a constant level. We have added this explanation in the text.

5. Page 6, line 8. The numbers of reactions are incorrect.

Corrected.

6. Page 6, lines 11-18. The reaction constants for these aqueous Hg reactions should be given either in the main text or in the supplement. Also, I speculate these reactions are also highly uncertain. Do the authors consider the uncertainties associated with them?

A table showing aqueous Hg reactions in our model was added as Table S1.

Aqueous Hg reduction is one of the major sources of GEM in the atmosphere. Therefore, aqueous Hg reactions is supposed to be a factor controlling GEM mixing ratios, which turns out to influence GOM mixing ratios. However, aqueous Hg reduction is not an important control on GOM simulations in this study because GEM mixing ratios in the model were fixed using observed values. The uncertainties associated with aqueous Hg reactions would not influence GEM mixing ratios and therefore have minor effects on simulated GOM mixing ratios.

7. Page 12, lines 1-9. These several sentences are confusing. At first, it is mentioned that "the patterns of diurnal variation were similar at the three sites". Then, it is said that "PM showed negligible diurnal variation". I suggest that a statistical method is used to quantitatively detect the diurnal patterns at all the sites.

We apologize for the confusion. All three sites did show diurnal cycles, the expression of "PM showed negligible diurnal variation" were intended to suggest the daily amplitude is very small compared to that at AI and TF. We have rephrased these sentences to clarify the point. The changed wording is as follows:

"The patterns of diurnal variation were similar at the three sites with small discrepancy on the occurring time of daily peaks (~ 13:00 LT at AI, and ~14:00 LT at TF and PM), but the magnitude varied largely by site. AI had the largest GOM diurnal amplitude (i.e., daily maximum – daily minimum) ranging from 0.73 to 13.29 ppqv, TF from 0.05 to 0.57 ppqv, and PM showed a very small range from 0.05 to 0.14 ppqv."

8. Table 2: How uncertain are the simulated [Br] at TF and PM? What is the major source of [Br]? How is the concentration of $Br_2$ set in the box model? In addition, are the boundary layer heights at AI and PM set to be constant? Do the authors expect any diurnal variations of the boundary layer height?

No observations of [Br] were available at the three sites. At AI, we used [BrO] observation from Saiz-Lopez et al. (2006) to constrain simulated [BrO]. However, at TF and PM, we don't have any data available to constrain Br species, so we did not give a specific source for Br and BrO. [$Br_2$] initial concentration was set to 1 pptv (e.g. Finley et al., 2008) but without setting it as constant. In result, $Br_2$ was rapidly depleted during daytime simulations with very low concentration of Br and BrO produced. Average daytime maximum of [BrO] is about 10 ppqv, and [Br] is negligible. The model simulation at TF and PM indicated that $O_3$ and OH were sufficient for GOM production at TF and PM.

The boundary layer heights at AI and PM were set to be constant. The major reason is we do not have diurnal cycle data of boundary layer height at AI and PM. Moreover, in the MBL, boundary layer height is usually a few hundred meters and does not vary much (Vickers and Mahrt, 2003; Angevine et al., 2006). At PM, the boundary layer height is set as averaged daytime boundary height at TF minus the elevation difference between the two sites. At night, due to its high elevation, it was above the nocturnal boundary layer.

9. Figures: The figures throughout the paper should use a consistent way of uncertainty quantification, probably being consistent with the statistical method used for the observations (Figure 1). In the current paper, min-max, standard deviation, and box-whiskers all exist making the readers difficult to compare the uncertainties among these figures.

We revised the figures and used error bars for standard deviation only.

In addition, I suggest the authors merge Figures 2, 3, and 8.

Thank you for the suggestion. We merged these three figures to Figure 2 in the revised manuscript. The merged figure was shown below (Fig. 1):

[Figure]

Figure 1. Average diurnal cycles of observed GEM (top panel) and simulated and observed GOM (bottom panel) averaged over the selected 50 days at Appledore Island (AI), 12 days at Thompson Farm (TF), and 21 days at Pack Monadnock ( PM) from summers of 2007, 2008, and 2010. The error bars represent standard deviation.

REFERENCE

Angevine, W. M., Hare, J. E., Fairall, C. W., Wolfe, D. E., Hill, R. J., Brewer, W. A. and White, A. B.: Structure and formation of the highly stable marine boundary layer over the Gulf of Maine, J. Geophys. Res., 111(D23), D23S22, doi:10.1029/2006JD007465, 2006.

Finley, B. D. and Saltzman, E. S.: Observations of $Cl_2$, $Br_2$, and $I_2$ in coastal marine air, J. Geophys. Res., 113(D21), D21301, doi:10.1029/2008JD010269, 2008.

Hogan, R. J., Grant, A. L. M., Illingworth, A. J., Pearson, G. N. and O'Connor, E. J.: Vertical velocity variance and skewness in clear and cloud-topped boundary layers as revealed by Doppler lidar, Q.J.R. Meteorol. Soc., 135(640), 635–643, doi:10.1002/qj.413, 2009.

Saiz-Lopez, A., Shillito, J. A., Coe, H., and Plane, J. M. C.: Measurements and modelling of $I_2$, IO, OIO, BrO and $NO_3$ in the mid-latitude marine boundary layer, Atmos. Chem. Phys., 6, 1513–1528, doi:10.5194/acp-6-1513-2006, 2006.

Vickers, D. and Mahrt, L.: Evaluating Formulations of Stable Boundary Layer Height, doi:10.1175/JAM2160.1, 2004.